# Multimodal single cell sequencing implicates chromatin accessibility and genetic background in diabetic kidney disease progression

Parker C. Wilson [1,6], Yoshiharu Muto [2,6], Haojia Wu [2], Anil Karihaloo[3], Sushrut S. Waikar [4] & Benjamin D. Humphreys [2,5] ✉

The proximal tubule is a key regulator of kidney function and glucose metabolism. Diabetic kidney disease leads to proximal tubule injury and changes in chromatin accessibility that modify the activity of transcription factors involved in glucose metabolism and inflammation. Here we use single nucleus RNA and ATAC sequencing to show that diabetic kidney disease leads to reduced accessibility of glucocorticoid receptor binding sites and an injury-associated expression signature in the proximal tubule. We hypothesize that chromatin accessibility is regulated by genetic background and closely-intertwined with metabolic memory, which pre-programs the proximal tubule to respond differently to external stimuli. Glucocorticoid excess has long been known to increase risk for type 2 diabetes, which raises the possibility that glucocorticoid receptor inhibition may mitigate the adverse metabolic effects of diabetic kidney disease.

Diabetes is the leading cause of end-stage renal disease (ESRD) and a significant contributor to morbidity and mortality[1]. An estimated 40% of patients with diabetes develop chronic kidney disease (CKD), which manifests as worsening proteinuria and renal dysfunction[2]. Single cell sequencing is a powerful technique that has advanced our understanding of kidney biology[3]. Multimodal integration of single nucleus RNA (snRNA-seq) and assay for transposase-accessible chromatin sequencing (snATAC-seq) provides insight into how transcription factors and chromatin-chromatin interactions regulate expression of nearby genes[4].

Glucocorticoids and mineralocorticoids comprise a class of hormones called corticosteroids produced in the adrenal cortex. Cortisol is the primary endogenous glucocorticoid that binds glucocorticoid receptor (GR), which is expressed in proximal tubule, thick ascending limb, endothelium, and podocytes[3]. Chronic exposure to endogenous cortisol and long-term treatment with synthetic glucocorticoids has been linked to type 2 diabetes and metabolic syndrome[5,6]. Single-cell sequencing of human DKD has shown that the thick ascending limb and distal nephron have a transcriptional signature consistent with altered corticosteroid signaling[7]. GR also acts on the proximal tubule where it increases expression of gluconeogenic genes to drive the synthesis of glucose from non-carbohydrate substrates like lactate, glutamine, and glycerol[8,9]. The kidney contributes approximately half of circulating glucose during prolonged fasting and studies have demonstrated that both glucose reabsorption and gluconeogenesis are increased in type 2 diabetes[8].

Genetic variation contributes to the risk of type 2 diabetes and DKD progression as shown by genome wide association studies

[1]Division of Anatomic and Molecular Pathology, Department of Pathology and Immunology, Washington University in St. Louis, St. Louis, MO, USA. [2]Division of Nephrology, Department of Medicine, Washington University in St. Louis, St. Louis, MO, USA. [3]Novo Nordisk Research Center Seattle Inc, Seattle, WA, USA. [4]Section of Nephrology, Department of Medicine, Boston University School of Medicine, Boston Medical Center, Boston, MA, USA. [5]Department of Developmental Biology, Washington University in St. Louis, St. Louis, MO, USA. [6]These authors contributed equally: Parker C. Wilson, Yoshiharu Muto. ✉e-mail: humphreysbd@wustl.edu

(GWAS)[10–12]. Many of the variants identified by GWAS are common (MAF > 0.01) and explain a small proportion of heritability of type 2 diabetes and kidney disease[12]. GWAS have been used to investigate a wide variety of kidney-function-related traits, however, one of the difficulties with GWAS is assigning function to risk variants located in non-coding regions[13,14]. Recent studies have shown that a significant proportion of GWAS variants for type 1 and type 2 diabetes are located in cell-specific open chromatin regions[15,16]. The relationship between gene expression and chromatin accessibility can be modeled using allele-specific chromatin accessibility (ASCA)[17–19]. snATAC-seq can measure ASCA by quantifying the ratio of ATAC peak fragments that intersect heterozygous germline SNV. It can also predict CRE gene targets within cis-coaccessibility networks (CCAN), which makes it a compelling method for estimating the effect of SNV in open chromatin regions[18]. These methods open the door to novel techniques for gene-enhancer predictions and quantitation of single cell allele-specific effects[20].

We have performed snRNA-seq and snATAC-seq on kidney cortex from patients with and without type 2 diabetes to identify cell-specific differentially expressed genes and accessible chromatin regions associated with diabetic kidney disease (DKD). We validated key findings from our multimodal analysis with cleavage under targets and release using nuclease (CUT&RUN) and CRISPR interference (CRISPRi) to directly measure transcription factor binding and modify chromatin accessibility of cis-regulatory elements (CRE). Epigenetic regulation of chromatin accessibility may contribute to long-term expression of DKD-related genes in a process termed metabolic memory[21]. Our analysis identified cell-specific changes in chromatin accessibility that co-localize with transcription factor binding sites associated with glucose metabolism and corticosteroid signaling in the diabetic nephron.

## Results

### Patient demographics and clinical information

For snRNA-seq and snATAC-seq, a total of thirteen kidney cortex samples were obtained from control patients ($n = 6$), and patients with diabetic kidney disease (DKD, $n = 7$). Tissue samples were collected following nephrectomy for renal mass ($n = 10$) or from deceased organ donors ($n = 3$). Patients ranged in age from 50 to 78 years (median = 57 y) and included seven men and six women (Supplementary Dataset 1). Patients with type 2 diabetes had elevated hemoglobin A1c (mean = 8.2 +/− 1.5%). The majority of patients with DKD were on antihypertensive or ACE inhibitor therapy and two patients were on insulin. Two patients with DKD had mild to moderate proteinuria as measured by urine dipstick.

### Renal histology of donor samples

Tissue sections were stained with H&E and examined by a renal pathologist (P.W.) to evaluate histological features of DKD. Control samples did not have significant global glomerulosclerosis (<10%) or interstitial fibrosis and tubular atrophy (<10%). Patients with DKD had predominantly mild ($N = 3$, <25%) or moderate ($N = 3$, 26–50%) global glomerulosclerosis with a corresponding increase in interstitial fibrosis and tubular atrophy (Supplementary Fig. 1). Mean eGFR of DKD samples (66 +/− 25 ml/min/1.73 m$^2$) and control samples (74 +/− 15 ml/min/1.73 m$^2$) was not statistically different (Students $t$-test, $p = 0.49$). DKD samples showed nodular mesangial expansion, thickened glomerular basement membranes and afferent arteriolar hyalinosis.

### Single nucleus ATAC sequencing in type 2 diabetes

The snATAC-seq dataset included six control samples and seven with DKD. snATAC-seq libraries were counted with cellranger-atac (10X Genomics) and aggregated prior to cell-specific peak calling with MACS2[20,22]. We detected 437,311 accessible chromatin regions ('ATAC peaks') across all cell types. More abundant cell types had a larger number of ATAC peaks compared to less common cell types, which is likely a function of increased power and sequencing depth (Supplementary Fig. 2). The aggregated dataset was analyzed in Signac following doublet removal with AMULET and batch effect correction with Harmony[20,23,24]. A total of 68,458 cells passed quality control filters and all major cell types in the kidney cortex were represented (Fig. 1A). Cell types were identified based on increased chromatin accessibility within gene body and promoter regions of lineage-specific markers (Supplementary Fig. 3) and enrichment for cell-specific ATAC peaks (Supplementary Dataset 2). The most abundant cell type was the proximal convoluted tubule (PCT), which comprised approximately one third of snATAC-seq cells, and DKD samples had a trend towards greater number of infiltrating leukocytes (mean of 42 vs. 211, $p = 0.10$), including B cells, T cells, and mononuclear cells. We previously described an injured population of VCAM1 + proximal tubule cells (PT_VCAM1) that increase in response to acute kidney injury, aging, and CKD[3]. PT_VCAM1 can be distinguished from PCT by expression of VCAM1 and HAVCR1 (KIM-1), which is a marker of kidney injury (Supplementary Fig. 3). There was a greater proportion of PT_VCAM1 in DKD samples compared to control samples (mean proportion 0.12 vs. 0.03, Wilcoxon rank sum $p = 0.004$), however, this proportion varied widely by donor (Supplementary Fig. 4). There was another closely related cluster of cells that we designated PT_PROM1 (Fig. 1A). The PT_PROM1 cluster was PROM1$^{high}$ VCAM1- in contrast to PT_VCAM1, which was PROM1$^{low}$ VCAM1+ (Supplementary Fig. 3). We hypothesize that the PT_PROM1 cluster represents a population of CD133 + VCAM1- cells that we previously identified in control kidney, which raises the possibility that there are multiple proximal tubule states related to cellular injury or inflammation[3].

### Cell-specific differentially accessible chromatin regions in the diabetic nephron

We compared DKD and control samples to identify 7358 cell-specific differentially accessible chromatin regions (DAR) that met the adjusted $p$-value threshold (Supplementary Dataset 3, Benjamini Hochberg $p$adj < 0.05), including 1315 that also met an absolute log-fold-change threshold of 0.1 (Fig. 1B). The majority of DAR showed decreased accessibility rather than increased accessibility (923 vs. 392) and many were located in a promoter region (Fig. 1B). In contrast, a minority of DAR were intergenic (156/1315, 11%) located a median distance of 22 kb from the nearest transcriptional start site (TSS). Approximately one third of intergenic DAR ($n = 59/156$, 37%) and one fourth of intronic DAR ($n = 58/240$, 24%) mapped to a FANTOM enhancer[25]. The proximal convoluted tubule (PCT) had the greatest number of DAR ($n = 422$) followed by the proximal straight tubule (PST), PT_VCAM1, and thick ascending limb (Fig. 1C). Less abundant cell types like podocytes and endothelial cells had few if any DAR, which likely reflects our limited power to detect them. Among 1315 total DAR, 975 were unique because a subset of DAR were shared between multiple cell types (Fig. 1C). DAR present in multiple cell types included regions within or near ATP1B1 (Supplementary Fig. 5 and 6). ATP1B1 encodes a subunit of the sodium potassium ATPase, suggesting that DAR in diabetes may elicit a conserved effect on ion transport across nephron segments.

We grouped DAR from the proximal convoluted tubule (PCT) and proximal straight tubule (PST) and annotated them with the nearest protein-coding gene to perform gene ontology enrichment. Genes near proximal tubule DAR were enriched for pathways involved in response to insulin, cellular response to peptides, response to hormone stimuli, and ion transport (Fig. 1D). Insulin resistance is a key feature of DKD and we identified proximal tubule DAR with decreased chromatin accessibility near multiple genes that regulate insulin signaling (Supplementary Dataset 3)[26]. There was a DAR in the second intron of the insulin receptor (INSR) that showed decreased accessibility in the proximal tubule (Fig. 1E, Orange Arrow). This region was predicted to regulate INSR expression via a CCAN (Fig. 1E, Green Arcs)

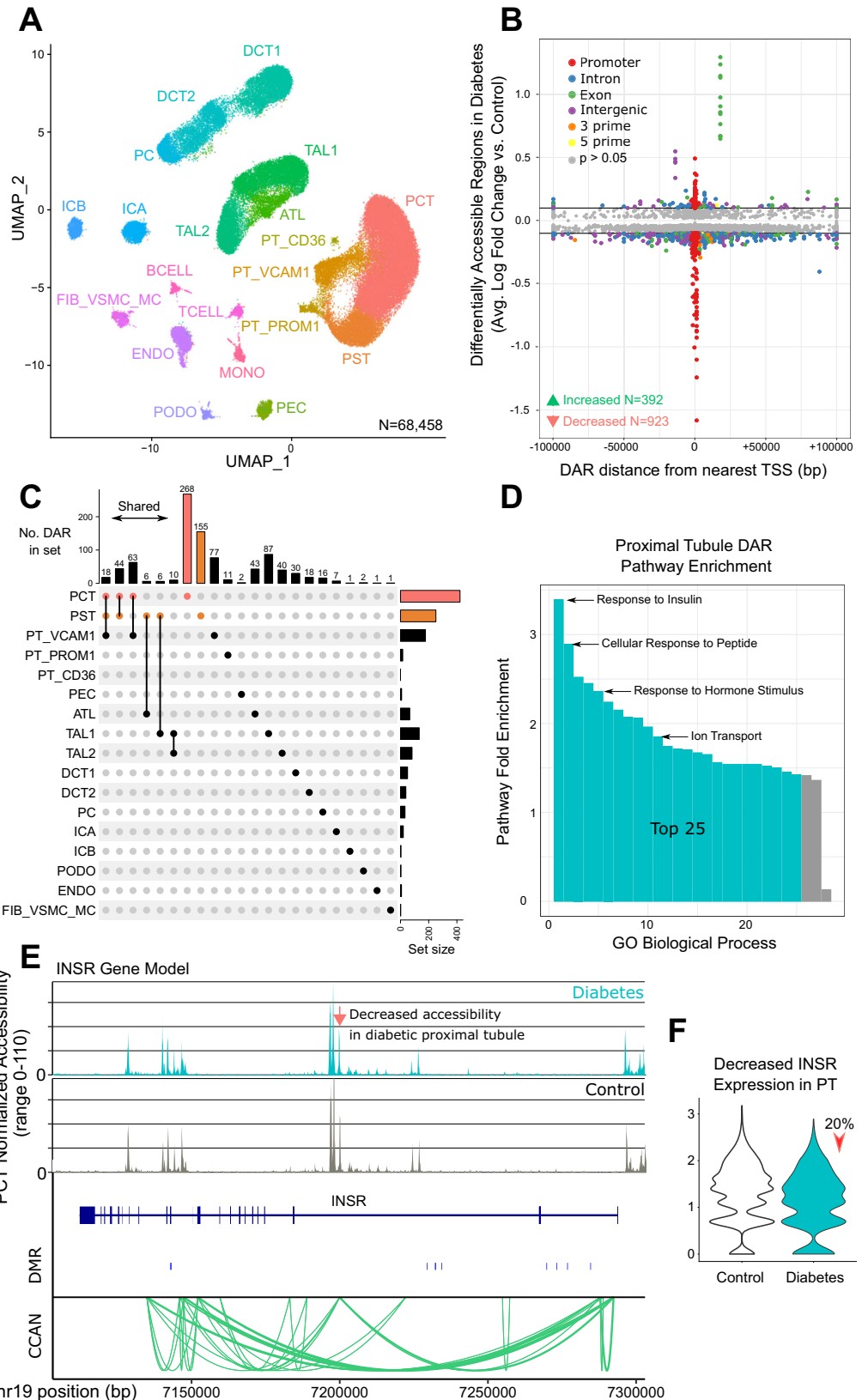

and was associated with decreased *INSR* expression in the corresponding snRNA-seq dataset (Fig. 1F). We did a pairwise comparison between all control and DKD snATAC-seq samples to show that the effect size and direction of this *INSR* DAR is largely reproducible across donors (Supplementary Fig. 7). The loop of Henle is a key regulator of sodium reabsorption where we identified two DAR in the promoter region of *ATP1B1* (Supplementary Fig. 6). These DAR were present in both the ascending thin limb (ATL) and thick ascending limb (TAL1, TAL2) where they showed decreased chromatin accessibility. Decreased chromatin accessibility was associated with decreased *ATP1B1* expression in the same cell types in the corresponding snRNA-seq dataset. Together, these findings suggest that DKD is associated

**Fig. 1 | snATAC-seq of human DKD. A** UMAP of snATAC-seq dataset. Six control and seven DKD samples with 68,458 cells. PCT-proximal convoluted tubule, PST-proximal straight tubule, PT_VCAM1-VCAM1(+) proximal tubule, PT_PROM1-PROM1(+) proximal tubule, PT_CD36-CD36(+) proximal tubule, PEC-parietal epithelial cells, ATL-ascending thin limb, TAL1-CLDN16(-) thick ascending limb, TAL2-CLDN16(+) thick ascending limb, DCT1-early distal convoluted tubule, DCT2-late distal convoluted tubule, PC-principal cells, ICA-type A intercalated cells, ICB-type B intercalated cells, PODO-podocytes, ENDO-endothelial cells, FIB_VSMC_MC-fibroblasts, vascular smooth muscle cells and mesangial cells, TCELL-T cells, BCELL-B cells, MONO-mononuclear cells. **B** Effect size and location of DAR in DKD. Control cell types were compared to DKD to identify cell-specific DAR (Source data are provided in Supplementary Dataset 3). Significance was evaluated with a Bonferroni-adjusted Wilcoxon Rank Sum test. DAR with padj < 0.05 that met an absolute log2-fold-change threshold of 0.1 (horizontal bars) were annotated relative to the nearest TSS. **C** DAR in DKD that are cell-specific or shared between cell types. DAR that were shared between multiple cell types or unique to a cell type are

displayed. (Source data are provided in Supplementary Dataset 3). **D** Proximal tubule DAR pathway enrichment. Cell-specific DAR from PCT and PST were annotated with the nearest protein-coding gene to perform gene ontology enrichment. Fold-enrichment for all significant GO biological processes is shown and the top 25 are highlighted (Source data are provided as a Source Data file). **E** Proximal tubule-specific DAR and ATAC peaks in the insulin receptor. snATAC-seq coverage plots for DKD and control PCT are displayed. The orange arrow indicates a DAR in intron 2 that shows decreased accessibility in DKD (chr19:7196798-7198626, fold-change = 0.92, $p$adj = 7.7 × $10^{-13}$). Differentially methylated regions (DMR) associated with end-stage kidney disease due to diabetes are shown as blue bars (see Methods). Green arcs depict the nodes of a cis-coaccessibility network (CCAN). Statistical significance was evaluated using a Bonferroni-adjusted Wilcoxon Rank Sum test. **F** Proximal tubule *INSR* expression by snRNA-seq. Control proximal tubule was compared to DKD to identify DEGs. DKD proximal tubule showed reduced *INSR* expression (fold-change = 0.78, $p$adj = 1.2 × $10^{-27}$). Statistical significance was evaluated using a Bonferroni-adjusted Wilcoxon Rank Sum test.

with changes in chromatin accessibility that regulate expression of genes important for insulin signaling and sodium reabsorption.

We compared the proximal convoluted tubule (PCT) and PT_VCAM1 to identify changes in chromatin accessibility associated with the pro-inflammatory PT_VCAM1 cell state (Supplementary Dataset 4). There were 4,498 DAR and the majority showed decreased accessibility ($N = 3055$, 68%). Gene ontology analysis of nearby protein-coding genes showed enrichment for pathways involved in kidney development, metabolism, amino acid transport, epithelial cell proliferation, response to glucocorticoids, and regulation of transforming growth factor beta signaling. ATAC peaks with increased chromatin accessibility in PT_VCAM1 were located near pro-inflammatory genes like *IL-6, CD40,* and *TGFB2* in addition to genes involved in proliferation like *EGFR* and *MYC*.

**Single nucleus RNA sequencing to detect differentially expressed genes in type 2 diabetes**

A total of eleven snRNA-seq libraries were aggregated with cellranger (10X Genomics) and analyzed with Seurat following doublet removal with DoubletFinder and batch effect correction with Harmony[4,23,27]. The snRNA-seq dataset included six control samples and five with DKD. A total of 39,176 cells passed quality control filters and all major cell types in the kidney cortex were represented (Fig. 2A), including the PT_VCAM1 population[3]. snRNA-seq cell types largely expressed the same lineage-specific markers that showed increased chromatin accessibility in the snATAC-seq dataset (Supplementary Fig. 8) and were enriched for cell-specific genes (Supplementary Dataset 5). There was a greater proportion of PT_VCAM1 in DKD compared to controls (mean proportion 0.06 vs. 0.02, Wilcoxon rank sum $p = 0.03$). We compared individual cell types between control and DKD samples to identify cell-specific differentially expressed genes (Supplementary Dataset 6). The cell type with the greatest number of differentially expressed genes was the proximal tubule ($N = 607$, $p$adj < 0.05, |avg_log2FC| > 0.25). Gene ontology analysis of differentially expressed genes in the proximal tubule showed significant overlap with snATAC-seq pathways, including membrane depolarization, anion homeostasis, sodium ion transport, and glucocorticoid signaling (Fig. 2B). The diabetic proximal tubule showed a modest increase in expression of GR (*NR3C1*, fold-change = 1.14, $p$adj = 4.7 × $10^{-10}$), although it did not meet the log-fold change threshold. The proximal tubule also showed increased expression of sodium glucose cotransporter 2 (SGLT2, fold-change = 1.24, $p$adj = 1.9 × $10^{-30}$) and increased expression of the rate-limiting enzyme in gluconeogenesis (*PCK1*, fold-change = 1.63, $p$adj = 1.8 × $10^{-41}$). Similar to our findings from the snATAC-seq analysis, a subset of differentially expressed genes were shared between multiple cell types (Fig. 2C). These shared genes were enriched for pathways involved in regulation of cell growth, cellular response to hypoxia, angiogenesis, cellular response to insulin stimulus, glucocorticoid

signaling, and ion transport. For example, *INSR* showed decreased expression in the proximal tubule (Fig. 1F), thick ascending limb, and distal convoluted tubule (Supplementary Dataset 6). Additional enzymes in the gluconeogenic pathway were also upregulated in the diabetic proximal tubule (Fig. 2D). Together, these findings suggest that the diabetic proximal tubule increases expression of genes that promote both glucose reabsorption (*SLC5A2*) and glucose production (*PCK1, ALDOB, FBP1, G6PC*). Comparison with the corresponding snATAC-seq dataset showed multiple proximal tubule DAR near *PCK1* (Fig. 2E, Orange Boxes), suggesting that changes in chromatin accessibility may lead to increased *PCK1* expression. *PCK1* DAR were located both within and distal to its gene body where they interacted with the promoter via a CCAN (Fig. 2E, Green Arcs). Similar relationships between PCT DAR and CCAN were observed near *ALDOB, FBP1*, and *G6PC* (Supplementary Figs. 9–11).

The diabetic thick ascending limb (TAL1) had 622 differentially expressed genes compared to controls ($p$adj < 0.05, |avg_log2FC| > 0.25). Differentially expressed genes in TAL1 were enriched for pathways involved in nitric oxide signaling, ATP biosynthesis, anion transport, and cellular response to cAMP, EGFR signaling, glucocorticoids, hypoxia, and insulin. Similar to the diabetic proximal tubule, there was decreased expression of *INSR* (fold-change = 0.76, $p$adj = 1.5 × $10^{-17}$) and increased expression of GR (*NR3C1*, fold-change = 1.26, padj = 4.7 × $10^{-10}$). There was also decreased expression of *HSD11B2* (fold-change = 0.71, $p$adj = 5.3 × $10^{-33}$), which is the enzyme that catalyzes the conversion of cortisol to the inactive metabolite cortisone to protect nonselective activation of MR. In fact, decreased *HSD11B2* expression was observed in every cell type in the distal nephron (Supplementary Dataset 6). These data support our hypothesis that the diabetic nephron has increased GR signaling due to increased GR expression and decreased activity of the enzyme responsible for metabolizing cortisol.

We compared the proximal tubule and PT_VCAM1 to identify differentially expressed genes associated with the PT_VCAM1 cell state. There were 3842 differentially expressed genes (Supplementary Dataset 7) enriched for pathways involved in cell migration, EGFR signaling, insulin receptor signaling, histone deacetylation, regulation of glycolysis, and TGF-beta signaling. For example, *INSR* expression was decreased in PT_VCAM1 relative to PT (fold-change = 0.67, $p$adj = 4.1 × $10^{-53}$) and *TGFBR2* was increased (fold-change = 1.34, $p$adj = 5.4 × $10^{-34}$). These changes were accompanied by a modest increase in GR expression (*NR3C1*, fold-change = 1.07, $p$adj = 2.0 × $10^{-10}$) and a marked reduction in *FKBP5* (fold-change = 0.45, $p$adj = 4.9 × $10^{-117}$).

**Cell-specific transcribed *cis*-regulatory elements**
Transcribed cis-regulatory elements (tCRE) confer cell type specificity and the majority are located in enhancer and promoter regions where

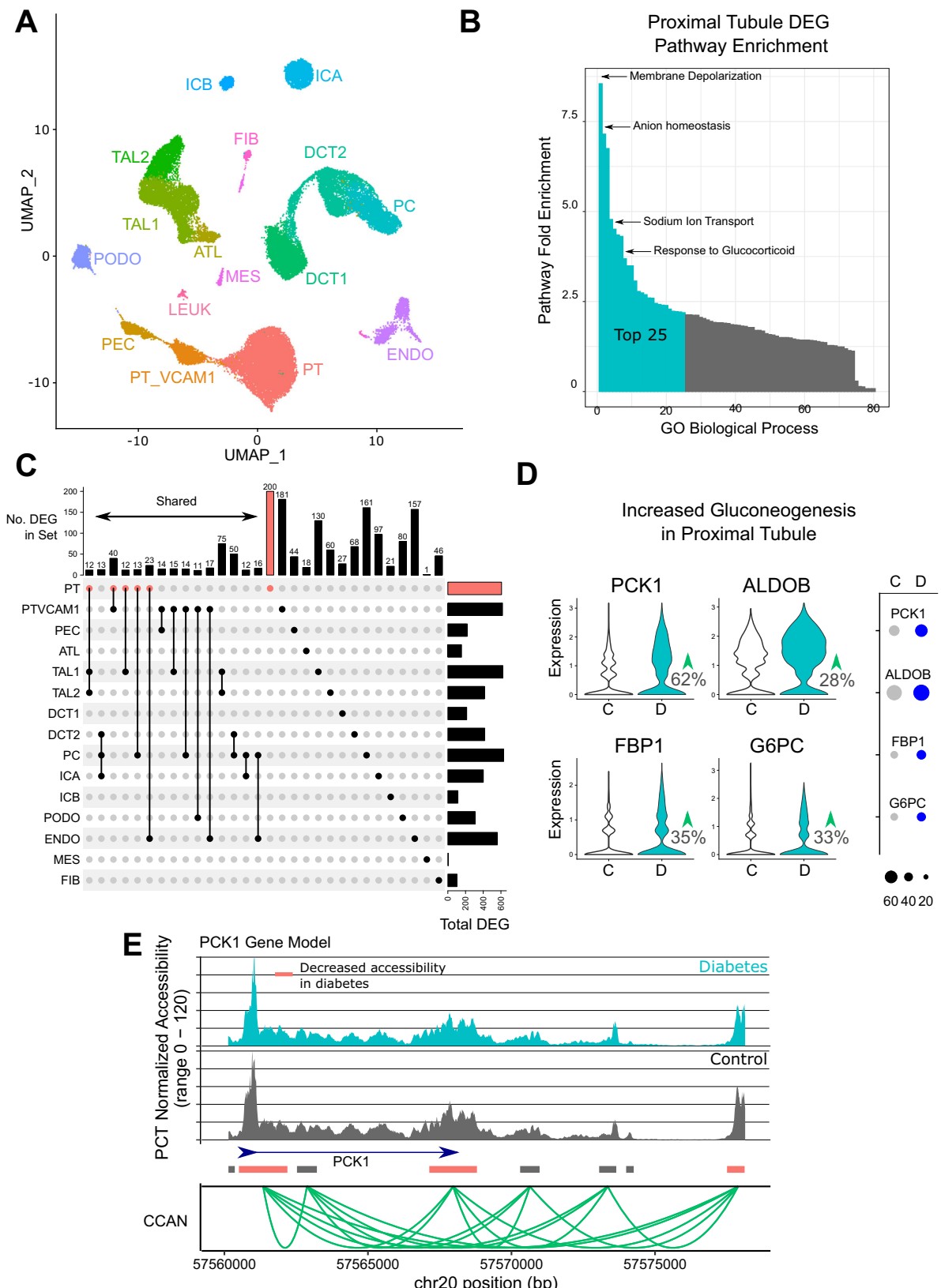

they overlap with ATAC peaks[28,29]. Transcriptional start sites (TSS) can be identified by 5′ RNA sequencing if read 1 is long enough (>81 bp) to include the junction between the template switch oligo (TSO) and TSS. This type of analysis is compatible with single cell 5′ paired-end chemistry (SC5P-PE, 10X Genomics), but will not work with libraries that only use read 2 for alignment (SC5P-R2, 10X Genomics). We

analyzed two control and two DKD snRNA-seq libraries with SC5P-PE sequencing to identify de novo transcriptional start sites in CRE using Single Cell Analysis of Five-prime Ends (SCAFE)[30,31]. SCAFE analyzes the 5′ end of RNA transcripts to identify reads mapping to the junction between the TSO and cDNA sequence to localize TSS within tCRE after filtering false positives with a logistic regression classifier. We

**Fig. 2 | snRNA-seq of human DKD. A** UMAP of snRNA-seq dataset. Six control and five DKD samples were aggregated, preprocessed, and filtered. A total of 39,176 cells are depicted. PT-proximal tubule, PT_VCAM1-VCAM1(+) proximal tubule, PEC-parietal epithelial cells, ATL-ascending thin limb, TAL1-CLDN16(-) thick ascending limb, TAL2-CLDN16(+) thick ascending limb, DCT1-early distal convoluted tubule, DCT2-late distal convoluted tubule, PC-principal cells, ICA-type A intercalated cells, ICB-type B intercalated cells, PODO-podocytes, ENDO-endothelial cells, MES-mesangial cells and vascular smooth muscle cells, FIB-fibroblasts, LEUK-leukocytes. **B** Proximal tubule DEG pathway enrichment. Significant cell-specific DEG from proximal tubule were used to perform gene ontology enrichment with Panther. Fold-enrichment for all significant GO biological processes is shown and the top 25 are highlighted (Source data are provided as a Source Data file). **C** DEG in DKD that are cell-specific or shared between cell types. DEG that were either shared between

multiple cell types or unique to a specific cell type are displayed. DEG shared between multiple cell types are limited to groups that share ten or more DEG (Source data are provided in Supplementary Dataset 6). **D** Proximal tubule shows increased expression of gluconeogenic genes by snRNA-seq. Control proximal tubule was compared to DKD proximal tubule in the snRNA-seq dataset to identify differentially expressed genes with the FindMarkers function and visualized as violin plots and dot plots. DKD proximal tubule showed increased expression of *PCK1, ALDOB, FBP1,* and *G6PC* (see Supplementary Dataset 6 for adjusted *p*-values). **E** Proximal tubule-specific DAR and ATAC peaks in *PCK1*. snATAC-seq coverage plots for DKD and control PCT are displayed in relation to the *PCK1* gene body. Orange bars indicate multiple DAR that show decreased accessibility in diabetic PCT (Supplementary Dataset 3). Green arcs depict the nodes of a cis-coaccessibility network (CCAN) surrounding the *PCK1* gene body.

---

identified 37,698 tCRE across all cell types (Fig. 3A, Supplementary Dataset 8). The majority of tCRE were near a protein-coding TSS (mean distance = 3823 +/− 43,885), but there was a significant proportion of tCRE in intronic (Fig. 3B, 11847/37,698, 31%) and intergenic regions (Fig. 3B, 1367/37,698, 3%). Some of these tCRE may represent enhancer RNA (eRNA), which are a family of non-coding RNA that regulate gene expression and enhancer activity in a cell-specific manner[32]. The majority of tCRE were overlapping with a snATAC-seq peak (Fig. 3C, 23,048/37,698, 61%) and approximately half of snATAC-seq DAR were overlapping with a tCRE (494/968, 51%, hypergeometric test $p = 3.6 \times 10^{-5}$). A small minority of tCRE were cell-type-specific ($N = 361/37,698$, 1%, Supplementary Dataset 8), but correspond to well-known cell-type-specific genes. For example, there was a cell-specific tCRE in podocytes in the promoter region of *NPHS1* and a cell-specific tCRE in the proximal tubule in the promoter region of *CUBN* (Supplementary Dataset 8). These data suggest that 5' snRNA-seq datasets contain complementary information that can be used to evaluate the activity of CRE identified by snATAC-seq.

We compared control to DKD samples to identify cell-specific differential tCRE in diabetes. Across all cell types, we detected a total of 293 differential tCRE (Supplementary Dataset 9). These tCRE included 139 unique regions, which were enriched for pathways involved in mitochondrial electron transport and angiogenesis. Multiple cell types showed increased transcription of CRE in promoters associated with oxidative phosphorylation like *MT-CO1, MT-CO2* and *MT-CO3*. We also compared the proximal tubule to PT_VCAM1 to identify tCRE that are enriched in the PT_VCAM1 cell state (Supplementary Dataset 10). Among 204 differential tCRE in PT_VCAM1, one of the most enriched tCRE was the *VCAM1* promoter (fold change = 1.46, *p*adj = $1.1 \times 10^{-82}$). The *VCAM1* promoter tCRE showed increased chromatin accessibility in the corresponding snATAC dataset (Fig. 3D) and was associated with two additional tCRE located ~60 kb upstream. Each of the upstream tCRE were near a snATAC-seq peak and linked to the *VCAM1* promoter via a CCAN (Fig. 3D). We previously reported that this upstream CRE binds NFkB by chromatin immunoprecipitation PCR[3]. NFkB signaling induces *VCAM1* expression in the proximal tubule, which raises the possibility that NFkB binding to the upstream CRE is also associated with transcription of enhancer RNA[32,33]. Together, these data suggest that single cell analysis of 5' ends may help to identify enhancers by prioritizing CRE that are actively transcribed.

## Glucocorticoid receptor CUT&RUN in bulk kidney cortex

Cellular response to glucocorticoids is influenced by pre-existing chromatin accessibility state where the majority of GR binding sites localize to open chromatin regions[34,35]. We used cleavage under targets and release using nuclease (CUT&RUN) to directly measure GR binding in bulk kidney cortex obtained from a control donor[36]. We identified 4362 GR binding sites (Supplementary Dataset 11) located in promoter regions ($N = 2889$, 66%), introns ($N = 744$, 17%) and distal intergenic regions ($N = 567$, 13%). The density of cell-specific ATAC peaks closely resembled the density of GR CUT&RUN sites across the genome, which

suggests that GR predominantly binds to areas of open chromatin in the kidney (Fig. 4A). GR binding sites overlapped with cell-specific ATAC peaks ($N = 3066$, 70%); many of which were shared between multiple cell types. We visualized the intersection between cell-specific ATAC peaks and CUT&RUN sites to identify individual cell types or groups of cells that share ten or more GR binding sites (Fig. 4B). The presence of GR binding sites within cell-specific ATAC peaks suggests GR signaling is controlled by chromatin accessibility and regulated by distinct GR modules shared across cell types. For example, there were GR binding sites in ATAC peaks unique to the proximal tubule ($N = 61$, Fig. 4B column 2), unique to the distal nephron ($N = 13$, Fig. 4B column 12), and shared between the proximal tubule and distal nephron ($N = 26$, Fig. 4B column 3). Similarly, there were GR binding sites unique to lymphocytes ($N = 15$, Fig. 4B column 13) and shared between the proximal tubule and lymphocytes ($N = 21$, Fig. 4B column 5).

## Transcription factor motif enrichment and activity in the diabetic nephron

We used the JASPAR database to identify over-represented transcription factor motifs in cell-specific ATAC peaks and DAR in DKD[37]. Cell-specific ATAC peaks were enriched for established transcription factors that drive cell type differentiation like HNF4A in the proximal tubule and TFAP2B in the distal nephron (Supplementary Dataset 12). Transcription factors that were enriched in DAR provide insight into cell-specific signaling pathways that are altered in DKD. PCT DAR were significantly enriched for NR3C1 and NR3C2 motifs (Supplementary Dataset 13, NR3C1 fold enrichment = 2.1, $p = 3.8 \times 10^{-259}$; NR3C2 fold enrichment = 2.1, $p = 2.4 \times 10^{-288}$). NR3C1 is the canonical binding motif for GR and NR3C2 is the binding motif for MR. The presence of NR3C1 and NR3C2 motifs within PCT DAR suggests that chromatin accessibility may regulate corticosteroid signaling in the diabetic proximal tubule[38,39]. We also saw enrichment of KLF9 and FOXO3 motifs within PCT DAR, which are downstream of GR activation (Supplementary Dataset 13). One of the most enriched motifs in PCT DAR was Histone H4 transcription factor (HINFP, fold enrichment = 6.9, $p = 6.3 \times 10^{-308}$). HINFP interacts with a component of the MeCP1 histone deacetylase complex (HDAC) involved in transcriptional repression, which may explain why the majority of DAR showed decreased chromatin accessibility[40]. To help prioritize active signaling pathways in DKD, we identified transcription factor motifs that were both differentially expressed and enriched in DAR (Fig. 5A). HIF1A showed increased expression and was enriched in DAR in multiple distal nephron cell types (PC, ICA, DCT2) and PT_VCAM1. HIF1A is hypoxia inducible factor 1 subunit alpha, a master regulator of cellular response to hypoxia in the kidney[41]. GR showed increased expression in the proximal tubule (PT) and thick ascending limb (TAL1) where NR3C1 motifs were also enriched in DAR. In contrast, MR showed decreased expression in the distal nephron (PC, DCT2), but increased expression in the thick ascending limb (TAL1). Together, these data suggest that corticosteroid signaling is altered in the diabetic proximal tubule and thick

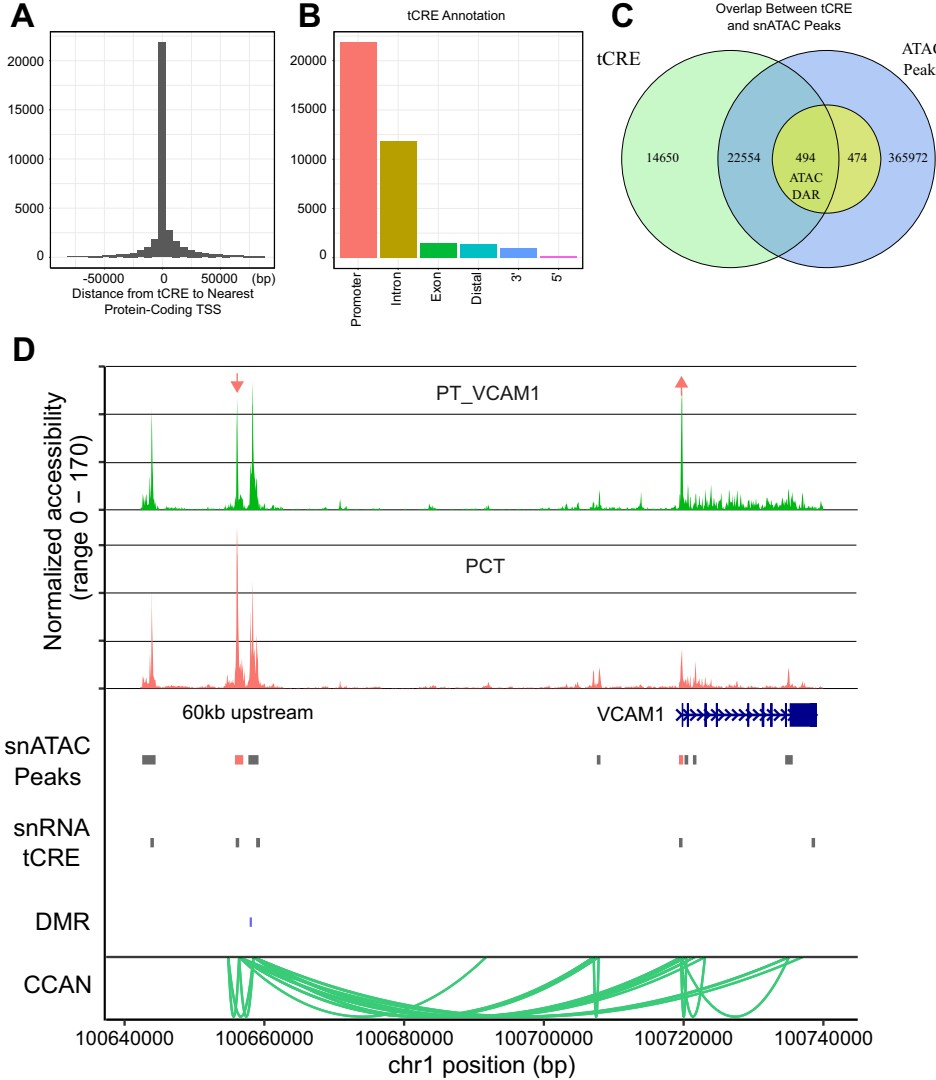

**Fig. 3 | Transcribed cis-regulatory elements (tCRE) detected by 5-prime snRNA-seq. A** Distance from tCRE to TSS. Two control and two DKD samples were sequenced with 5′ paired-end sequencing and analyzed with SCAFE. A total of 37,698 tCRE were annotated with ChIPSeeker and displayed relative to the nearest TSS (Source data is provided as a Source Data file). **B** Annotation of tCRE. The relative proportion of tCRE in promoters, introns, exons, distal intergenic, 3-prime, and 5-prime regions is shown (Source data is provided as a Source Data file). **C** Overlap between tCRE and ATAC peaks. tCRE were intersected with cell-specific ATAC peaks and DAR in DKD using GenomicRanges. **D** Proximal convoluted tubule and PT_VCAM1 DAR and ATAC peaks in *VCAM1*. snATAC-seq coverage plots for PT_VCAM1 (green) and PCT (orange) are displayed in relation to the *VCAM1* gene body. The orange arrows indicate DAR that show either increased or decreased accessibility in PT_VCAM1 relative to PCT (Supplementary Dataset 4). snATAC-seq peaks accessible in the proximal tubule are displayed (snATAC peaks, gray boxes) in the same track as PCT DAR (snATAC peaks, orange boxes). snRNA-seq tCRE regions are displayed below snATAC-seq peaks and DAR (snRNA tCRE, gray boxes). Differentially methylated regions (DMR) in publicly available databases associated with end-stage kidney disease due to diabetes are shown as blue bars (see Methods). Green arcs depict the nodes of a cis-coaccessibility network (CCAN) surrounding the *VCAM1* gene body.

---

ascending limb where multiple cell types may be exposed to a hypoxic environment.

We used chromVAR to compare transcription factor activity between control and diabetic cell types (Supplementary Dataset 14). chromVAR is a tool for inferring transcription-factor-associated chromatin accessibility in single cells that helps to address sparsity inherent in snATAC-seq datasets[42]. This is a qualitatively different and unbiased approach when compared to transcription factor motif enrichment analysis because it is not limited to a pre-specified list of cell-specific DAR. Diabetic PCT showed decreased transcription factor activity for NR3C1 (Fig. 5B, fold-change = 0.56, $p$adj = $1.1 \times 10^{-70}$) and increased activity for REL motifs (Fig. 5B, fold-change = 2.08, $p$adj = $6.9 \times 10^{-212}$), which was a pattern observed throughout the nephron. These data support our transcription factor motif enrichment analysis by further demonstrating that NR3C1 motifs localize to areas of decreased chromatin accessibility in diabetes and REL motifs localize to areas of increased accessibility.

### Glucocorticoid receptor footprinting with snATAC-seq and CUT&RUN in RPTEC

We used the snATAC-seq dataset to perform transcription factor footprinting for GR to visualize the relationship between NR3C1 motifs and chromatin accessibility. Across all cell types, there was a well-defined footprint immediately surrounding NR3C1 motifs (Fig. 5C). DKD samples showed reduced chromatin accessibility surrounding NR3C1 motifs, however, this effect was attenuated when we limited our analysis to the proximal tubule (Fig. 5C). We cultured immortalized renal proximal tubule epithelial cells (hTERT-RPTEC, ATCC) and performed CUT&RUN to find 22,539 consensus GR binding sites that were not present in IgG-stimulated

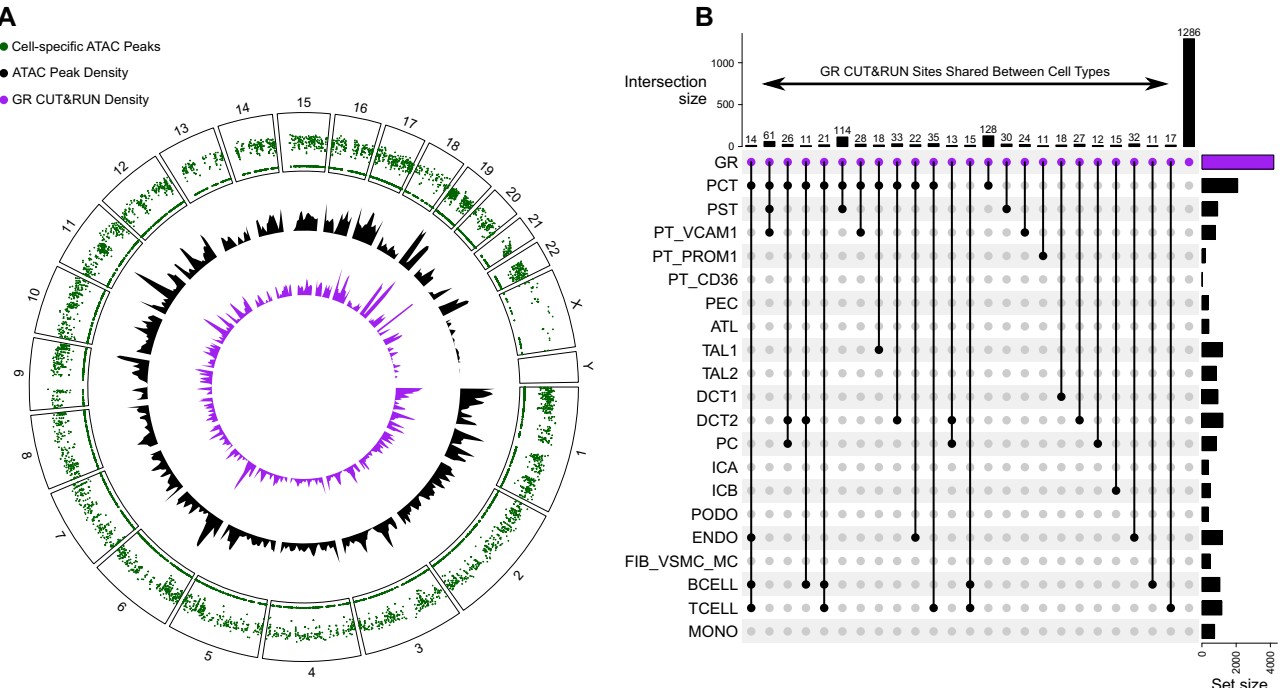

**Fig. 4 | Glucocorticoid receptor (GR) CUT&RUN in bulk kidney cortex. A** Density of GR CUT&RUN sites relative to cell-specific ATAC peaks. Cell-specific ATAC peaks were identified with the Seurat FindMarkers function (Supplementary Dataset 2) and converted into a rainfall plot (green track) using the circlize package in R. Each dot in the rainfall plot corresponds to a cell-specific ATAC peak and the y-axis corresponds to the log-transformed minimal distance between the peak and its two neighboring peaks. Clusters of peaks appear as a "rainfall" in the plot. The density of cell-specific ATAC peaks (black track) and GR CUT&RUN peaks (purple track) are shown adjacent to the rainfall plot using a 10 Mb default window size. **B** Upset plot showing intersection between GR CUT&RUN sites and cell-specific ATAC peaks. Each column of the Upset plot indicates a unique grouping of ATAC peaks that intersect with bulk kidney GR CUT&RUN sites. The solid black circles in each column indicate which cell types are present within the intersection at the top of the plot. For example, the first column is a group of 14 ATAC peaks shared between PCT, ENDO, B-cells, and T-cells that each contain a GR CUT&RUN site that is not seen in other cell types. Only intersections with ten or more GR CUT&RUN sites are included in the plot. GR CUT&RUN sites that do not intersect a cell-specific ATAC peak are displayed as the black bar to the far right (N = 1296). The horizontal bars on the right of the Upset plot represent the total number of peaks within each cell type or GR CUT&RUN group (Source data is provided in Supplementary Dataset 2 and 11).

negative control samples (Supplementary Dataset 15). hTERT-RPTEC media is supplemented with 25 ng/ml hydrocortisone (i.e., cortisol) and is a model of long-term glucocorticoid exposure. Nearly half of PCT DAR (168/422, 39%) were overlapping with a GR CUT&RUN site. These findings are comparable to the chromVAR analysis, which showed that ~47% of control PCT and 35% of diabetic PCT ATAC peaks contain an NR3C1 motif. GR binding sites that do not directly overlap a PCT DAR may interact with DAR via CCAN (Fig. 5D). For example, we identified multiple GR binding sites throughout the *FKBP5* gene body located in promoter and intronic regions (Fig. 5E, Purple Boxes). Some of the GR binding sites in *FKBP5* co-localized with PCT DAR (Fig. 5E, Orange Box), but others did not. *FKBP5* expression was decreased throughout the entire nephron (Fig. 5F), which highlights its potential importance in DKD and raises the possibility that changes in chromatin accessibility regulate its expression. The high proportion of overlap between GR binding sites and snATAC-seq PCT DAR was especially striking given that hTERT-RPTEC are a cell culture model that does not fully recapitulate the normal proximal tubule. We profiled open chromatin regions in hTERT-RPTEC and primary RPTEC using Omni-ATAC and compared them to the PCT snATAC-seq dataset. Approximately 59% of hTERT-RPTEC ATAC peaks (N = 57,675/ 96,162, Supplementary Dataset 16) and 56% of primary RPTEC ATAC peaks (N = 80,322/141,198, Supplementary Dataset 17) were overlapping with cell-specific PCT snATAC-seq peaks. These data suggest that hTERT-RPTEC and primary RPTEC capture roughly half of the chromatin accessibility profile of a normal proximal tubule cell.

## Validation of differentially expressed genes in a bulk RNA-seq dataset of human DKD

We analyzed a previously published bulk RNA-seq dataset of human DKD to determine if our snRNA-seq findings are broadly generalizable. The dataset published by Fan et. al consisted of 9 controls, 6 with early DKD, and 22 with advanced DKD[43]. Early DKD was defined as eGFR > 90 mL/min/1.73 m² and UACR < 300 mg/g. Advanced DKD was defined as eGFR < 90 mL/min/1.73 m² or UACR > 300 mg/g. According to this definition, samples from our study would be categorized as advanced DKD because all of them had either eGFR < 90 mL/min/1.73 m² or proteinuria. Another important difference between our study and the study by Fan et al. is that mean eGFR of control samples from our study (66 +/− 25 ml/min/1.73 m²) was significantly less than eGFR of control samples from their study (87 +/− 9.8 ml/min/1.73 m², p = 0.03).

We compared the transcriptional profile of advanced DKD to control samples from Fan et al. to identify 9,632 differentially expressed genes (Supplementary Dataset 18A, BH padj < 0.05). Roughly half of these differentially expressed genes were upregulated (N = 5181) and the remaining were downregulated (N = 4451). These differentially expressed genes were enriched for familiar pathways including amino acid metabolism, B cell receptor signaling, T cell differentiation, response to tumor necrosis factor, response to peptide hormone, cellular response to hormone stimulus, and ion transmembrane transport. The enrichment of pathways involved in lymphocyte signaling and differentiation likely reflects the greater proportion of leukocytes present in DKD samples compared to control samples. We previously demonstrated that advanced DKD samples from Fan et al. contain an increased proportion of leukocytes and PT_VCAM1[3].

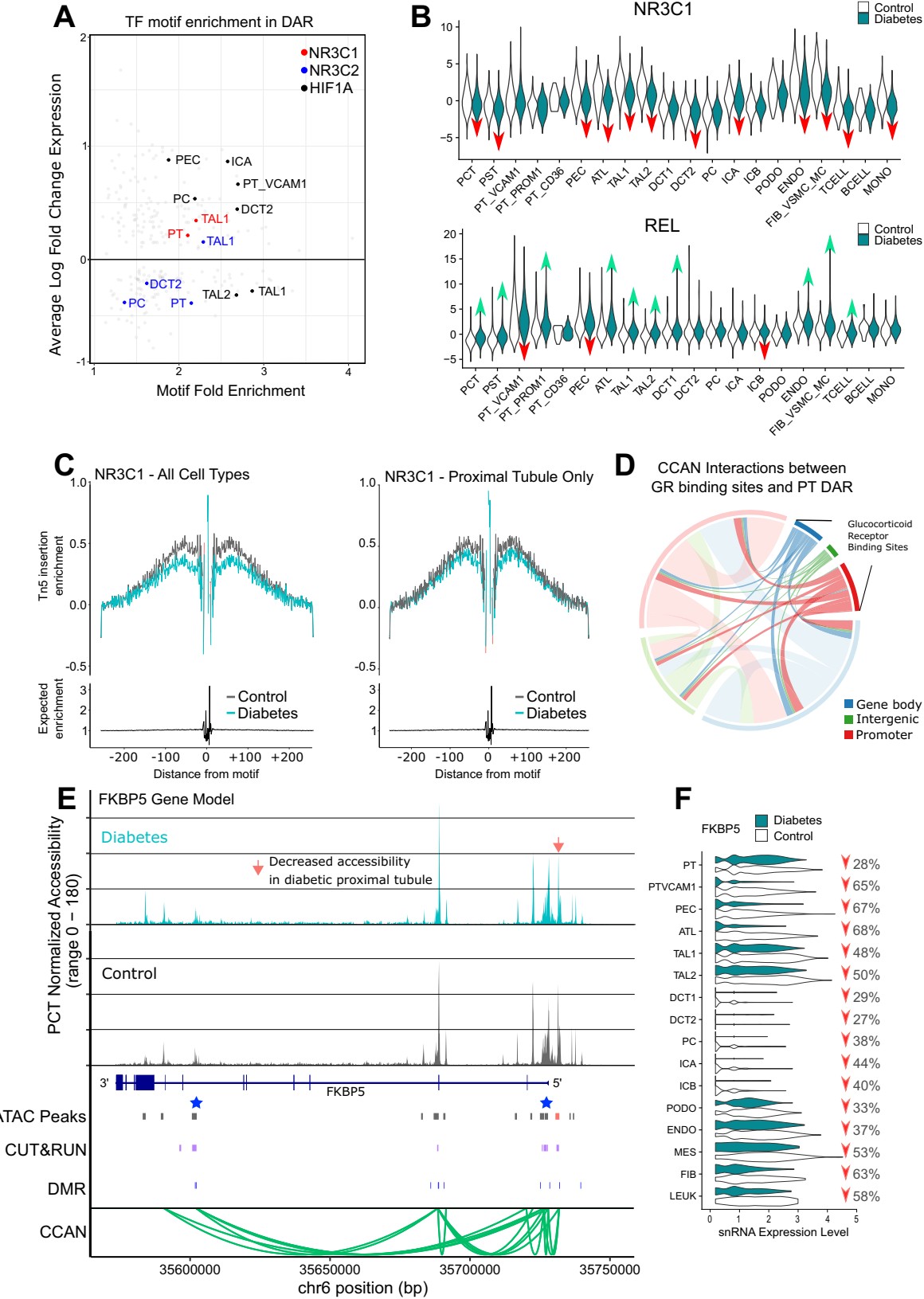

Advanced DKD samples showed increased expression of GR (*NR3C1*, fold-change = 1.16, *p*adj = 0.02) and *VCAM1* (fold-change = 1.36, *p*adj = 0.006) and decreased expression of *INSR* (fold-change = 0.49, *p*adj = $1.6 \times 10^{-12}$), *HSD11B2* (fold-change = 0.38, *p*adj = $3.6 \times 10^{-8}$) and *FKBP5* (fold-change = 0.46, *p*adj = 0.0009). Next, we compared early DKD samples to controls to identify 1041 differentially expressed genes,

among which 385 were upregulated and 656 were downregulated (Supplementary Dataset 18B). The differentially expressed genes in early DKD were enriched for pathways involved in response to epidermal growth factor, cellular response to glucocorticoids, cellular response to insulin stimulus, and cellular response to tumor necrosis factor. In contrast to the advanced DKD samples, we did not detect

**Fig. 5 | Glucocorticoid receptor (GR) binding and *FKBP5* in DKD. A** Cell-specific transcription factor expression and motif enrichment. Transcription factors that were both differentially expressed (Supplementary Dataset 6) and showed motif enrichment in cell-specific DAR (Supplementary Dataset 13) were visualized. Cell types that showed differential expression and motif enrichment for GR (NR3C1, red), MR (NR3C2, blue), and HIF1A (HIF1A, black) motifs are highlighted. PT-proximal tubule, PT_VCAM1-*VCAM1*(+) proximal tubule cells, PEC-parietal epithelial cells, TAL1-*CLDN16*(-) thick ascending limb, TAL2-*CLDN16*(+) thick ascending limb, DCT2-late distal convoluted tubule, PC-principal cells, ICA-type A intercalated cells. **B** Cell-specific chromVAR motif activity for GR and REL. chromVAR was used to compute cell-specific activities for NR3C1 and REL motifs for control and DKD. Red arrows indicate significantly decreased motif activity and green arrows indicate increased activity (see Supplementary Dataset 14). **C** Transcription factor footprinting for GR. Transcription factor footprinting analysis was performed for NR3C1 (GR) for all cell types and for PCT only to quantitate Tn5 insertion enrichment. **D** Interaction between PCT DAR and hTERT-RPTEC GR CUT&RUN sites. PCT

DAR in DKD (Supplementary Dataset 3) were intersected with cis-coaccessibility networks (CCAN) to identify all CCAN links that contain at least one PCT DAR. These regions were intersected with hTERT-RPTEC GR CUT&RUN sites and visualized with the circlize package in R to identify links between GR CUT&RUN sites and PCT DAR. **E** PCT-specific DAR and ATAC peaks in *FKBP5*. snATAC-seq coverage plots for DKD and control PCT are displayed in relation to *FKBP5*. The orange arrow indicates a DAR that shows decreased accessibility in DKD (Supplementary Dataset 3). PCT-specific ATAC peaks (Peaks, dark gray boxes) and DAR (Peaks, orange box) are shown in relation to hTERT-RPTEC CUT&RUN sites (GR, purple boxes), differentially methylated regions (DMR) associated with end-stage kidney disease due to diabetes (see Methods), and a cis-coaccessibility network (CCAN, green arcs) surrounding *FKBP5*. Blue stars indicate sites targeted by CRISPRi. **F** Cell-specific expression of *FKBP5* by snRNA-seq. Individual cell types were compared between control and DKD and visualized to display relative change in *FKBP5* expression. Red arrows indicate decreased *FKBP5* expression (see Supplementary Dataset 6 for adjusted *p*-values).

differential expression of *NR3C1, INSR, HSD11B2*, or *FKBP5* in early DKD samples. This difference may reflect reduced specificity of bulk RNA-seq and a limited number of early DKD samples (*n* = 6) vs. advanced DKD samples (*n* = 22), or alternatively, that these genes are associated with DKD progression.

### Differentially methylated regions in DKD and CKD overlap with cell-specific DAR and GR binding sites

We compiled a list of differentially methylated regions (DMR) from previously published studies that compared methylation patterns between DKD or CKD and control kidney samples[44–48]. Collectively, these studies identified DMR associated with kidney disease progression that we compared to our cell-specific DAR and GR binding sites. Approximately 9% of cell-specific DAR (*N* = 120/1315, 9.1%) were located within 1 kb of a DMR associated with ESKD due to diabetes, eGFR decline in DKD, or interstitial fibrosis in CKD and/or DKD (Supplementary Dataset 19). This overlap included 645 unique DMR enriched for pathways involved in lipid homeostasis, amino acid transport, ion transport, inflammatory response, and cellular response to hormone stimuli. There were multiple DMR located within or near the *INSR* (Fig. 1E), gluconeogenic genes (Supplementary Figs. 9–11), and *FKBP5* (Fig. 5E), including DMR in the *FKBP5* promoter that overlapped with a PCT cell-specific DAR and GR binding site (Fig. 5E: orange arrow). All of the *FKBP5* DMR showed increased methylation in association with DKD and *FKBP5* was identified as a top-ranked gene with multiple differentially methylated CpGs supported by functional data and additional cohorts[45,48]. Another notable DMR overlapped a cell-specific DAR in the *ATP1B1* promoter with a nearby a GR binding site (Supplementary Figs. 5–6). A similar proportion of GR CUT&RUN peaks in bulk kidney (*N* = 269/4362, 6,1%) and hTERT-RPTEC (*N* = 1537/22517, 6.8%) overlapped with DMR (Supplementary Dataset 19). Together these data support the hypothesis that increased methylation in DKD and/or CKD leads to decreased chromatin accessibility in key regulatory regions, including GR binding sites near *FKBP5*.

### CRISPRi knockdown of FKBP5 *cis*-regulatory elements

We selected two GR binding sites in the *FKBP5* gene body (Fig. 5E, Blue Stars) located at nodes within a CCAN (Fig. 5E, Green Arcs) to target with CRISPRi. These GR binding sites intersected with DMR in publicly available databases and a DAR in the *FKBP5* promoter. Catalytically inactive dCas9 fused to the Krüppel-associated box (KRAB) repression domain (dCas9-KRAB) reduces chromatin accessibility to induce targeted gene silencing. We transduced primary RPTEC with sgRNA targeting the TSS or intronic CRE in *FKBP5* to repress chromatin accessibility with dCas9-KRAB (Fig. 6A)[49]. Transduction of sgRNAs targeting the TSS or intronic region induced a 30–50% reduction in *FKBP5* expression compared to non-targeting control sgRNA (Fig. 6B). Furthermore, gene silencing was specific to *FKBP5* because CRISPRi did

not affect expression of neighboring genes expressed in primary RPTEC (Fig. 6C). We hypothesize that CRISPRi simulates the effect of hypermethylation, which leads to reduced chromatin accessibility.

### Partitioned heritability of Cell-specific ATAC peaks and differentially accessible regions for kidney-function-related GWAS traits

GWAS have shown that a growing list of kidney-related traits have a genetic component[50–53]. We downloaded GWAS summary statistics for eGFR, CKD, microalbuminuria, and urinary sodium excretion to determine whether cell-specific chromatin accessibility patterns explain heritability of these traits. First, we partitioned heritability of cell-specific ATAC peaks with stratified linkage disequilibrium score regression to prioritize which cell types explain heritability of kidney-function-related traits after controlling for baseline enrichment[54]. The cell types that showed the greatest enrichment for heritability of eGFR after correction for multiple comparisons were segments of the proximal tubule (PCT, PST) and the PT_VCAM1 population (Fig. 7A). This relationship between proximal tubule and heritability of eGFR has been previously described[55]. Interestingly, PT_VCAM1 cell-specific peaks also showed increased heritability for CKD, which raises the possibility that genetic background may influence the transition from proximal tubule to PT_VCAM1 (Fig. 7A). Multiple segments of the thick ascending limb (TAL1, TAL2) and principal cells (PC) showed enrichment for urinary sodium excretion, which is consistent with their known roles in sodium reabsorption. In contrast, we did not identify any cell types that showed increased heritability for microalbuminuria. This may reflect our reduced sensitivity to detect podocyte-specific ATAC peaks that may regulate this phenotype. Next, we partitioned heritability for cell-specific DAR that change in DKD. Similar to the findings from our cell-specific ATAC peak analysis, the DAR in the proximal tubule showed increased heritability of eGFR (Fig. 7B). In addition, DAR in the thick ascending limb (TAL1) showed increased heritability of urine sodium excretion (Fig. 7B). These data suggest that DKD induces changes in chromatin accessibility in some of the same regions that predict heritability of cell-specific kidney functions. It also raises the possibility that genetic background may modulate chromatin accessibility patterns to influence changes in eGFR or sodium excretion in DKD.

### Allele-specific chromatin accessibility as a modifier of gene expression

We created an open-source and containerized workflow for single-cell allele-specific analysis called "SALSA ([https://github.com/p4rkerw/SALSA])". SALSA is a tool for genotyping, phasing, mapping bias correction, and modeling of single-cell allele-specific counts obtained from snRNA-seq or snATAC-seq datasets. SALSA was developed from an earlier pipeline that used direct genotyping of snRNA-seq and

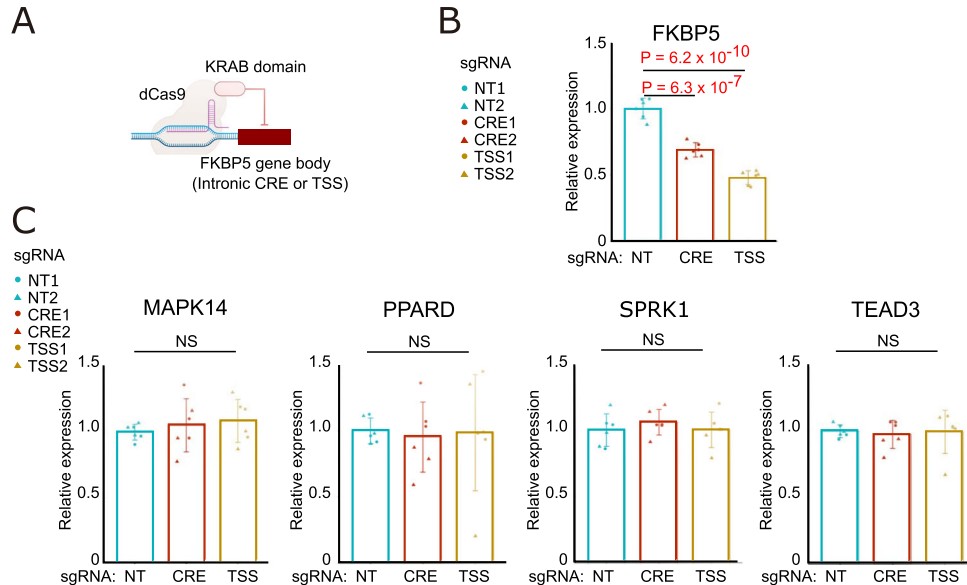

**Fig. 6 | Knockdown of *FKBP5* cis-regulatory elements with CRISPR interference.** **A** CRISPR interference diagram. dCas9-KRAB domain fusion protein and small guide RNAs (sgRNA) were used to target the TSS and a potential intronic CRE in the *FKBP5* gene. Targeted regions are depicted as blue stars in the *FKBP5* gene model diagram in Fig. 5E. sgRNA primers and region coordinates are provided in Supplementary Dataset 21. **B**, **C** Quantitative PCR of CRISPRi. RT and real-time PCR analysis of mRNAs for *FKBP5* and surrounding genes (*MAPK14, PPARD, SPRK1* and *TEAD3*) in primary renal proximal tubular epithelial cells (primary RPTEC) with CRISPR interference targeting the TSS and predicted cis-regulatory element (CRE) for *FKBP5*. NT, non-targeting control. Each group consists of $n = 2$ biologically independent experiments each with $n = 3$ biological replicates (2 sgRNAs with 3 biological replicates). Bar graphs represent the mean and error bars are the s.d. *p*-values are calculated with one-way ANOVA and a post-hoc Dunnett's test for multiple comparisons. Statistical significance was evaluated as an adjusted *p*-value < 0.05 (Source data are provided as a Source Data file).

snATAC-seq to explore allele-specific expression[3]. It has been updated to include reference-based variant phasing, multithreading, generalized linear models for ASCA, user-friendly tutorials and a publicly available Docker container built on the Genome Analysis Toolkit (GATK) developed by the Broad Institute[56]. SALSA uses GATK best practices for germline short-variant discovery to identify SNV and indels, which are phased using shapeit4 and a population-based reference from 1000 Genomes[57,58]. Phased variants present in the population-based reference are used to perform mapping bias correction and eliminate technical artifacts with WASP[59]. Heterozygous germline SNV that overlap ATAC peaks identify single-cell allele-specific peak fragments that map to either the reference or alternate allele. In this manner, heterozygous SNV in ATAC peaks are used as markers to assign a peak fragment to one haplotype or the other. This is an attractive approach because the reference haplotype can serve as a perfectly matched internal control for each individual. We quantitated single-cell allele-specific peak fragments in the proximal tubule and plotted the aggregate ratio of fragments mapping to reference or alternate alleles among 43,479 peaks containing a heterozygous SNV (Fig. 7C). The majority of peaks had an equal proportion of fragments mapping to each allele ($N = 35,019$, 80%), but a minority of peaks had allelic bias as evaluated by a binomial test ($N = 8460$, 20%). A significantly smaller proportion of peaks met the adjusted *p*-value threshold ($N = 542$, 1.2%), suggesting that most proximal tubule peaks do not show allelic bias when aggregated across a population. Next, we integrated snRNA-seq and snATAC-seq datasets to apply an algorithm developed by Ma et al. to identify ATAC peaks that are correlated with expression of nearby genes after correction for distance, GC content, peak accessibility, and peak width[60]. This approach helped to identify one or more gene targets for each peak containing a heterozygous SNV. We developed a simple mixed effect logistic regression model where the binary dependent variable was coded as the presence of an alternate allele in an ATAC peak fragment and the continuous predictor variable was single cell target gene expression in the integrated multimodal dataset. A mixed effect per sample was included to control

for pseudo-replication bias[61]. Our approach is a modification of a previously published model in SnapATAC used to identify gene-enhancer pairs that coded the dependent variable as 'open' or 'closed' and omitted the mixed effect[20]. Our base model evaluates whether increased or decreased expression of a target gene is predictive of the presence of an alternate allele within an ATAC peak. In the simplest terms, we can ask if the presence of a SNV in an ATAC peak is associated with changes in gene expression.

In the base model, we evaluated 66,828 peak-gene combinations to estimate the effect of gene expression on the presence of a heterozygous SNV in an ATAC peak (Fig. 7D, Supplementary Dataset 20A). The peak-gene combinations included 42,990 unique ATAC peaks where the majority had either one ($N = 28,989$, 67%) or two gene targets ($N = 8767$, 20%). Approximately 11% of peak-gene combinations showed nominal evidence of an allele-specific effect (Fig. 7D, 7512/66,828, Wald test *p* < 0.05), which decreased to 1% after adjustment for multiple comparisons ($N = 714$, 1%). There were 5,908 unique ATAC peaks with at least one nominally significant peak-gene allelic effect, predominantly in promoter ($N = 2312$, 39%) and intronic regions ($N = 2082$, 35%). The number of peak-gene combinations that showed increased expression in association with an alternate allele ($N = 3664$, 48%) was similar to the number of combinations that showed increased expression in association with the reference allele ($N = 3848$, 52%). Among peaks that met the significance threshold, the median absolute coefficient value was 0.01 (log odds) for a 1% increase in target gene expression. For a 10% increase in gene expression, this translates to the typical ATAC peak being 1.10 times more likely to contain an alternate allele in the base model. It is important to note that the base model may overestimate the effect of expression on allele-specific chromatin accessibility because it does not account for the role of diabetes. In subsequent models, we added a fixed effect for diabetes (Model 2) and a fixed effect for diabetes with an interaction term between target gene expression and diabetes (Model 3). Model 2 had 7557 peak-gene combinations where expression was a nominally significant predictor of the presence of an alternate allele (Supplementary

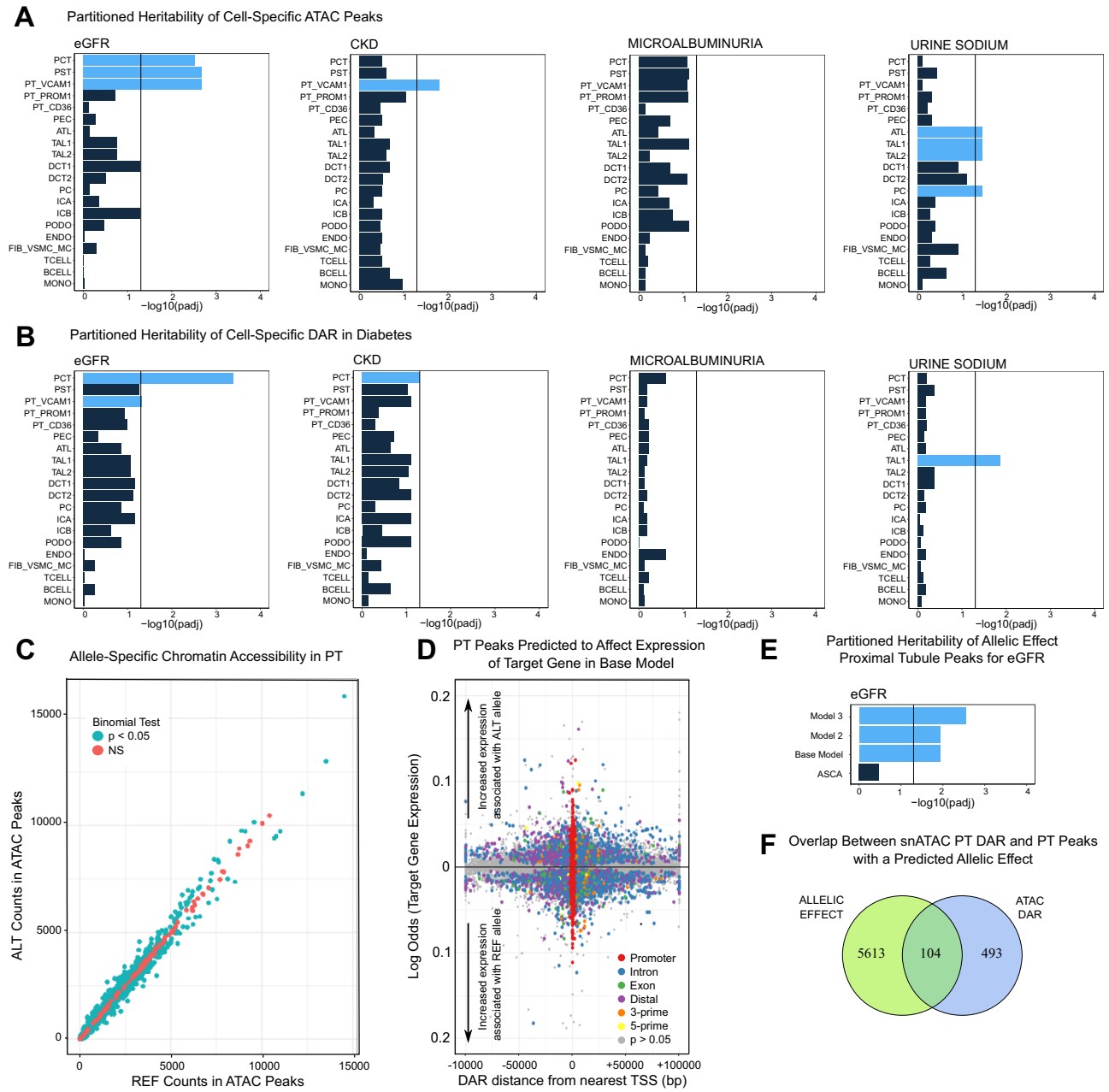

**Fig. 7 | Partitioned heritability of GWAS traits and predicted allelic effects with SALSA. A** Cell-specific analysis. Cell-type-specific ATAC peaks (Supplementary Dataset 2) were partitioned for heritability of GWAS traits using the ldsc cell-type-specific workflow. Significance was evaluated with a Benjamini-Hochberg-adjusted one-sided test using $p$adj < 0.05. $N$ = 13 biologically independent samples containing 68,458 cells were examined in a joint analysis (Source data are provided as a Source Data file). **B** Cell-specific DAR that change in DKD. Cell-specific DAR in DKD (Supplementary Dataset 3) were analyzed with ldsc using the cell-type-specific workflow. Significance was evaluated as described above. **C** Ratio of snATAC-seq fragments in the proximal tubule. SALSA was used to identify heterozygous SNV in the proximal tubule (PCT, PST) and counts mapping to the reference or alternate allele were aggregated across libraries and evaluated for allele-specific chromatin accessibility using an exact binomial test (see Supplementary Dataset 20. **D** Predicting an allele-specific effect with SALSA. The presence of a fragment mapping to an alternate allele in a proximal tubule peak (binary dependent variable) was

modeled as a function of target gene expression (continuous predictor variable) after controlling for sample-to-sample variability with a mixed effect per library using glmer in lme4. Effect size is displayed in log-odds where a 1 unit increase corresponds to a 1% increase in gene expression. $N$ = 11 biologically independent samples containing 26,929 proximal tubule cells were examined in a joint analysis (see Supplementary Dataset 20). **E** Partitioned heritability of proximal tubule peaks with a predicted effect. Peaks that were associated with changes in target gene expression were partitioned for heritability of eGFR for each of three generalized linear mixed models in addition to peaks that met the binomial threshold for allele-specific chromatin accessibility. Significance was evaluated with a Benjamini-Hochberg-adjusted one-sided test using $p$adj < 0.05. $N$ = 11 biologically independent samples were examined in a joint analysis (Source data are provided as a Source Data file). **F** Overlap between proximal tubule DAR and peaks with a predicted effect. DAR from PCT and PST were intersected with proximal tubule peaks with a predicted effect in Model 3.

Dataset 20B, Wald test $p < 0.05$), which included 6980 that were also identified in the base model. Since we know that diabetes can alter gene expression, Model 3 included an interaction term between expression and diabetes. Model 3 had 7353 peak-gene combinations

where expression was a nominally significant predictor of the presence of an alternate allele (Supplementary Dataset 20C, Wald test $p < 0.05$), which included 4577 that were also identified in the base model. Approximately 28% of these peak-gene combinations ($N$ = 2097/7353)

had a nominally significant interaction between expression and diabetes. In addition, diabetes was a nominally significant predictor of the presence of an alternate allele in 20% of peak-gene combinations ($N = 1471/7353$) after adjusting for gene expression. To examine the reproducibility of allele-specific effects, we analyzed peak-gene combinations in the proximal tubule for individual donors using the base model, but omitting the mixed effect. Across donors, we identified 24,794 nominally significant peak-gene combinations that were present in at least one donor. Approximately half of these peak-gene combinations ($n = 10,495$, 42%) were nominally significant in two or more donors. Two-thirds of the peak-gene combinations that were identified in multiple donors had an effect size in the same direction ($n = 7121/10,495$, 67%), suggesting that the presence of a variant in an ATAC peak may have similar effects across individuals.

We hypothesized that peaks with a predicted allele-specific effect would be enriched for heritability of kidney-function-related traits in the proximal tubule. We partitioned heritability for eGFR using peaks that met the nominal $p$-value threshold (Wald test $p < 0.05$) for the expression fixed effect in each of three models. All three models showed increased heritability for eGFR (Fig. 7E). In contrast, proximal tubule ATAC peaks that showed increased proportion of fragments mapping to the alternate or reference allele (Fig. 7C) in the aggregated dataset did not have enrichment for heritability of eGFR (ASCA, Fig. 7D). These peaks do not necessarily have a predicted allele-specific effect and may represent a random subset of proximal tubule peaks that exhibit biased chromatin accessibility due to chance alone. These data suggest that peaks with a predicted allele-specific effect are more likely to contribute to heritability of eGFR than a random sampling of proximal tubule peaks. Approximately 20% of proximal tubule DAR were overlapping with a peak with a predicted allelic effect ($N = 104/597$, Fig. 7E, Hypergeometric $p = 1.7 \times 10^{-10}$), suggesting that genetic background may also modify chromatin accessibility patterns in DKD. We used the 4476 significant peak-gene combinations present in all three models to perform gene ontology enrichment. The most enriched pathways involved peptide antigen assembly with MHC class II, antigen processing, and immunoglobulin production. Each of these pathways involve multiple HLA genes, which are known to exhibit allele-specific expression due to genetic variation in CRE[62]. Proximal tubule expression of MHC class II regulates the response to kidney injury and renal fibrosis[63]. Additional enriched pathways with important function in the proximal tubule included triglyceride metabolism, amino acid transport, and carbohydrate metabolism.

## Discussion

DKD progression is multifactorial and contributing factors include hyperglycemia, hypertension, hypoxia, and inflammation[64]. These factors exert their effect on different cell types throughout the nephron, which organizes a coordinated response to tissue injury. DKD was associated with an increased proportion of *VCAM1* + proximal tubule cells (PT_VCAM1) and infiltrating leukocytes in both snRNA-seq and snATAC-seq datasets. The PT_VCAM1 cell state emerges after proximal tubule injury and is associated with acute kidney injury, aging, and DKD[3,65]. It adopts a pro-inflammatory phenotype characterized by enhanced NFκB signaling and failed repair that may underlie transition from acute kidney injury to CKD[66]. In our dataset, there was another closely related proximal tubule cluster that we termed PT_PROM1. PT_PROM1 is characterized by a *PROM1*high *VCAM1*- chromatin accessibility profile that differentiates it from PT_VCAM1, which is *PROM1*low *VCAM1*+. These data are consistent with recent reports that there are multiple proximal tubule injury states and raises the question whether they have variable effects on kidney disease progression[67]. These cell states are not specific to DKD because they are observed in control kidney, however, we hypothesize that DKD leads to an increase in their abundance.

GR signaling is a key regulator of the immune response and exerts its effects on multiple cell types in the kidney. GR has potent anti-inflammatory properties that help mitigate tissue injury, but long-term exposure to glucocorticoids can lead to insulin resistance and metabolic syndrome[68]. Chromatin accessibility pre-determines cellular response to glucocorticoids and GR preferentially binds areas of open chromatin[35]. Our snATAC-seq analysis showed that the majority of DAR in DKD had reduced chromatin accessibility and were enriched for GR motifs across multiple cell types. These data suggest that the diabetic nephron is pre-programmed to respond differently to corticosteroids. Metabolic memory is an epigenetic state characterized by persistent expression of DKD-related genes despite glycemic control[21]. Decreased chromatin accessibility of GR binding sites within GR-responsive genes can lead to reduced transactivation and expression of target genes, however, DNA-binding-independent mechanisms may remain intact. GR directly binds pro-inflammatory transcription factors like NFκB to inhibit their activity in a process called tethering[69]. In a simple model, the metabolic effects of GR signaling can be attributed to transactivation and the anti-inflammatory effects can be attributed to tethering[70,71]. We hypothesize that the diabetic kidney adapts to a pro-inflammatory environment by remodeling chromatin accessibility to promote anti-inflammatory effects of GR at the expense of its adverse effects on metabolism. Targeting GR signaling in the proximal tubule may help to decrease GR-mediated gluconeogenesis and improve glycemic control, particularly during fasting. SGLT2i have been shown to increase gluconeogenesis, which raises the possibility that GR inhibition may be useful as a combination therapy, however, further studies will be needed to evaluate this hypothesis[70,72,73].

We used CUT&RUN to identify GR binding sites in the proximal tubule and validate predictions from our snATAC-seq analysis. GR binding sites showed significant overlap with proximal-tubule-specific ATAC peaks and participated in CCAN with cell-specific DAR in diabetes. A subset of GR CUT&RUN sites showed reduced chromatin accessibility in the proximal tubule, suggesting that it may respond differently to glucocorticoids. Changes in GR signaling were compounded by increased expression of GR, and reduced expression of *FKBP5* and *HSD11B2* in diabetes. *FKBP5* is a cytosolic chaperone that negatively regulates GR signaling as part of a negative feedback loop and *HSD11B2* converts cortisol into inactive cortisone to protect non-selective activation of MR[74]. We found DAR within *FKBP5* that coincide with GR binding sites within a CCAN. CRISPRi targeting of GR binding sites decreased *FKBP5* expression, suggesting that DKD is associated with reduced activity of GR negative feedback in the proximal tubule. *FKBP5* methylation has been associated with type 2 diabetes and cardiovascular risk and *FKBP5* polymorphisms are associated with insulin resistance[75,76]. Publicly available datasets of differentially methylated regions (DMR) associated with CKD and/or DKD progression overlapped with cell-specific DAR and GR CUT&RUN sites from our study. These data support the hypothesis that *FKBP5* hypermethylation leads to reduced chromatin accessibility in GRE and reduced activity of the GR negative feedback loop (Fig. 8).

The diabetic proximal tubule had increased gluconeogenesis, which is downstream of GR signaling. The proximal tubule is the primary site in the kidney for glucose production and its rate-limiting enzyme is *PCK1*[77]. We saw increased expression of *PCK1* and other gluconeogenic enzymes in the diabetic proximal tubule that was associated with reduced expression of *INSR*. Glucose reabsorption and glucose production are closely intertwined and tightly regulated by insulin signaling[78]. Proximal-tubule-specific *INSR* knockout and proximal-tubule-specific *IRS1/2* knockout have both been shown to increase gluconeogenesis, which is normally suppressed by insulin signaling or glucose reabsorption via SGLT2[78,79]. Glutamine is the preferred substrate for gluconeogenesis in the proximal tubule, which leads to ammonia production and acid excretion to maintain acid-base

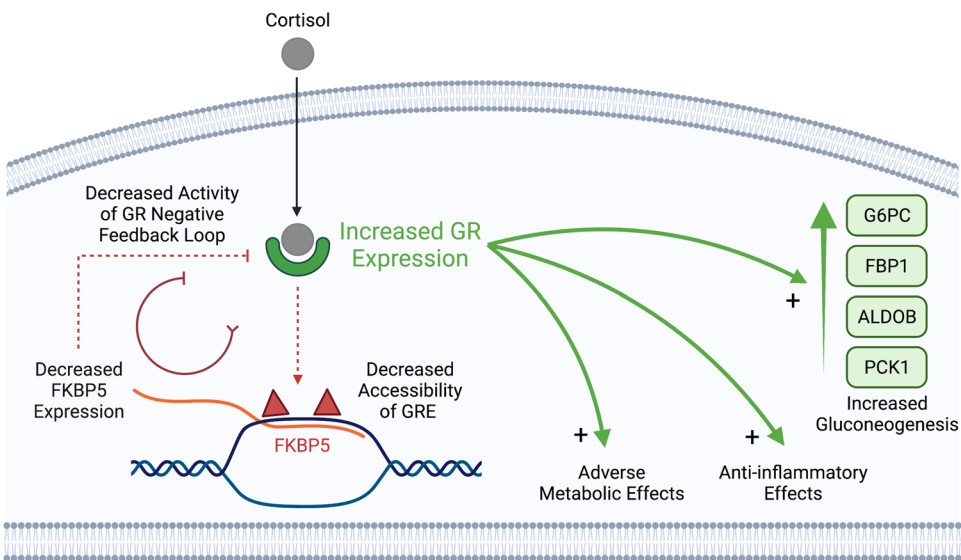

**Fig. 8 | Model of altered glucocorticoid receptor signaling in the diabetic proximal tubule.** GR expression is increased in the diabetic proximal tubule. Cortisol binds GR and translocates to the nucleus where it localizes to glucocorticoid response elements (GRE) in genes like *FKBP5*. Decreased chromatin accessibility of GRE in the *FKBP5* gene body is observed as reduced accessibility of proximal-tubule-specific ATAC peaks in DKD (red triangles). Reduced accessibility of *FKBP5* GRE leads to reduced transactivation by GR and reduced *FKBP5* expression. Reduced *FKBP5* expression decreases activity of the GR negative feedback loop. In the absence of *FKBP5* negative feedback, GR can exert both DNA-binding-dependent and DNA-binding-independent actions that lead to adverse metabolic effects, anti-inflammatory effects, and increased gluconeogenesis. GR hallmark pathway genes that are differentially expressed in our study include: *CDKN1A, CREBBP, EGR1, FKBP5, FOS, HSP9OAA1, ICAM1, JUN, MAPK10, NCOA2, NFKB1, NR3C1, NR4A1, PBX1, PCK2, PRKACB, SGK1, SMARCA4, STAT1, STAT5A, STAT5B.*

balance during prolonged fasting. It is possible that gluconeogenesis is upregulated in DKD to promote acid excretion and mitigate metabolic acidosis[80]. SGLT2i stimulate gluconeogenesis in the liver and kidney, however, it remains unclear whether gluconeogenesis affects DKD progression[72,73,78].

Genetic background is increasingly recognized as an important determinant of kidney function and DKD[10,50,53]. GWAS are generating a growing list of variants associated with kidney function, but it remains difficult to associate these variants with regulation of a specific gene or pathway. Bioinformatics approaches have led to the identification of quantitative trait loci associated with expression (eQTL), chromatin accessibility (caQTL), methylation (meQTL), and other traits that may regulate kidney function in a cell-specific manner[14]. Some of these phenotypes are driven by allele-specific effects that can be measured as changes in expression (ASE) and chromatin accessibility (ASCA)[3,19,81]. Allele-specific analysis can substantially boost the power of QTL studies because each individual serves as its own perfectly matched control. Multimodal single-cell datasets can take advantage of this approach to model allele-specific effects as a function of gene expression (or any other measurable quantity). We developed a tool for single cell allele-specific analysis called SALSA and used it to detect proximal-tubule-specific ATAC peaks in CRE that modify gene expression via ASCA. These peaks were enriched for heritability of eGFR and some of them coincide with DAR in DKD. These findings raise the possibility that genetic background affects kidney function via ASCA, which could alter the progression of DKD.

## Methods

### Human kidney tissue

This research complies with all relevant ethical regulations and has been approved by the Washington University Institutional Review Board. For snRNA-seq and snATAC-seq, non-tumor kidney cortex samples ($n = 10$) were obtained from patients undergoing partial or radical nephrectomy for renal mass at Brigham and Women's Hospital (Boston, MA) under an established Institutional Review Board protocol approved by the Mass General Brigham Human Research Committee.

An additional three kidney cortex samples (1 control and 2 DKD) were obtained from deceased organ donors in the Novo Nordisk biorepository. For bulk kidney GR CUT&RUN and Omni-ATAC, kidney cortex samples were obtained from deceased organ donors ($N = 3$) under an established Institutional Review Board protocol approved by Washington University in St. Louis. All participants provided written informed consent in accordance with the Declaration of Helsinki, including publication of demographic and clinical history as included in Supplementary Dataset 1. Histologic sections were reviewed by a renal pathologist and laboratory data was abstracted from the medical record.

### Statistics and reproducibility

Statistical analysis was conducted on all collected samples and data and carried out in Docker containers to enhance reproducibility. The details of each analysis is outlined in the methods section. No statistical method was used to predetermine sample size. No data were excluded from the analyses. The experiments were not randomized. Investigators were not blinded to allocation during experiments and outcome assessment.

### Nuclear dissociation and library preparation

Samples were chopped into <2 mm pieces, homogenized with a Dounce homogenizer (885302−0002; Kimble Chase) in 2 ml of ice-cold Nuclei EZ Lysis buffer (NUC-101; Sigma-Aldrich) supplemented with protease inhibitor (5892791001; Roche) with or without RNase inhibitors (Promega, N2615 and Life Technologies, AM2696, only for snRNA-seq library preparation), and incubated on ice for 5 min. The homogenate was filtered through a 40-μm cell strainer (43−50040−51; pluriSelect) and centrifuged at 500 × *g* for 5 min at 4 °C. The pellet was resuspended, washed with 4 ml of buffer, and incubated on ice for 5 min. Following centrifugation, the pellet was resuspended in Nuclei Buffer (10 × Genomics, PN-2000153) for snATAC-seq, nuclei suspension buffer (1× PBS, 1% bovine serum albumin [BSA], 0.1% RNase inhibitor) for snRNA-seq, or 1× PBS containing 1% BSA for CUT&RUN. The suspension was then filtered through a 5-μm cell strainer (43-50005-03, pluriSelect) and counted.

## Single nucleus ATAC sequencing and bioinformatics workflow

Thirteen snATAC-seq libraries were created with 10X Genomics Chromium Single Cell ATAC v1 chemistry following nuclear dissociation. These libraries included six control and seven DKD samples. Five of the control snATAC-seq libraries were prepared for a prior study (GSE151302). A target of 10,000 nuclei were loaded onto each lane. Sample index PCR was performed at 12 cycles. Libraries were sequenced on an Illumina Novaseq instrument, demultiplexed with bcl2fastq, and counted with cellranger-atac v2.0 (10X Genomics) using GRCh38. Libraries were aggregated with cellranger-atac without depth normalization. A mean of 327,328,680 reads were sequenced for each snATAC library (s.d. = 47,171,305) corresponding to a median of 15,150 fragments per cell (s.d. = 3875). The mean fraction of reads with a valid barcode was 96.3 ± 2.2%. Aggregated datasets were processed with Seurat v4.0.3 and its companion package Signac v1.3.0. A Seurat object was created using the CreateSeuratObject function with min.cells=10 and min.features=200. Low-quality cells were removed from the aggregated snATAC-seq dataset (peak region fragments > 2500, peak region fragments < 20,000, nucleosome signal < 4, TSS enrichment > 2) before normalization with term-frequency inverse-document-frequency (TFIDF) with default parameters. Dimensional reduction was performed via singular value decomposition (SVD) of the TFIDF matrix. Batch effect was corrected with Harmony using the RunHarmony function in Seurat and the lsi reduction. Dimensional reduction was performed with the FindNeighbors function using dimensions 2:30 and clustered using the FindClusters function with the Louvain algorithm. UMAP was performed using the harmony reduction and dimensions 2:30. Homotypic and heterotypic doublets were identified by running AMULET (v1.1.0) on individual snATAC-seq libraries and visualized in the aggregated object prior to removal of doublets with a $q$val < 0.05[24]. A gene activity matrix was constructed by counting ATAC peaks within the gene body and 2 kb upstream of the transcriptional start site using protein-coding genes annotated in the Ensembl database. The gene activity matrix was log-normalized prior to label transfer with the aggregated snRNA-seq Seurat object using canonical correlation analysis. The aggregated snATAC-seq object was filtered using label transfer to remove additional heterotypic doublets not captured by AMULET. Cell-specific ATAC peaks were called with MACS2 (v2.2.7.1) using the Signac wrapper with default parameters and a new Seurat object was created using MACS2 peaks and the FeatureMatrix function. The new snATAC-seq object was reprocessed with TFIDF, SVD, and batch effect correction followed by clustering and annotation based on lineage-specific gene activity as previously described. After filtering, there was a mean of 6000 ± 1134 nuclei per snATAC-seq library with a mean of 8098 ± 3231 peaks detected per nucleus. The final snATAC-seq library contained a total of 437,311 unique peak regions among 68,458 nuclei and represented all major cell types within the kidney cortex. Representative quality control plots and visualization of sample integration are in Supplementary Fig. 12. Differential chromatin accessibility between cell types was assessed with the Signac FindMarkers function for peaks detected in at least 20% of cells using a likelihood ratio test. Bonferroni-adjusted $p$-values were used to determine significance at an FDR < 0.05. Genomic regions containing snATAC-seq peaks were annotated with ChIPSeeker (v1.5.1) and clusterProfiler (v4.0.5) using Ensembl and FANTOM databases on hg38. Motif enrichment within DAR was calculated with the Signac FindMotifs function using cell-specific accessible peaks matched for GC content with TFBSTools (v1.39.0) and motifmatchr (1.14.0). chromVAR (1.14.0) motif activities were computed using the Signac wrapper and JASPAR2020 database (v0.99.10) adjusted for the number of fragments in peaks for each nucleus. CCAN were computed with cicero (v1.3.4.11) using the run_cicero function and default parameters. Gene-enhancer links were computed with the Signac Link-Peaks function and imputed RNA following label transfer and integration of snRNA-seq and snATAC-seq datasets.

## Single nucleus RNA sequencing and bioinformatics workflow

Eleven snRNA-seq libraries were obtained using 10X Genomics Chromium single cell chemistry following nuclear dissociation. Eight snRNA-seq libraries (5 control, 3 DKD) were prepared for prior studies (GSE131882, GSE151302). A target of 10,000 nuclei were loaded onto each lane. The cDNA for snRNA libraries was amplified for 17 cycles. Libraries were sequenced on an Illumina Novaseq instrument, demultiplexed with bcl2fastq, and counted with cellranger v4.0 using a custom pre-mRNA GTF built on GRCh38 to include intronic reads. Datasets were aggregated with cellranger v4.0 without depth normalization. A mean of 382,207,065 reads (s.d. = 78,522,614) were sequenced for each snRNA library corresponding to a mean of 68,429 reads per cell (s.d. = 21,706). The mean sequencing saturation was 77.6 ± 11.9%. The mean fraction of reads with a valid barcode (fraction of reads in cells) was 75.9 ± 6.6%. Aggregated datasets were pre-processed with Seurat v4.0.3 to remove low-quality nuclei (Features > 500, Features < 5000, RNA count < 16000, %Mitochondrial genes < 0.5, %Ribosomal protein large or small subunits < 0.3) and DoubletFinder v2.0.3 to remove heterotypic doublets (assuming 6% of barcodes represent doublets). The filtered library was normalized with SCTransform using default parameters and corrected for batch effects with Harmony v0.1.0 using the RunHarmony function in Seurat with the SCT assay. After filtering, there was a mean of 3561 ± 2028 cells per snRNA-seq library and a mean of 2137 ± 1031 genes detected per nucleus. Representative quality control plots and visualization of sample integration are in Supplementary Fig. 13. Clustering was performed by constructing a KNN graph with the Louvain algorithm using dimensions 1:24 and a resolution of 0.8. Dimensional reduction was performed with the RunUMAP function using dimensions 1:24 and individual clusters were annotated based on the expression of lineage-specific markers. The final snRNA-seq library contained 39,176 cells and represented all major cell types within the kidney cortex. Differential expression between cell types was assessed with the Seurat FindMarkers function for transcripts detected in at least 20% of cells. Bonferroni-adjusted $p$-values were used to determine significance at an FDR < 0.05.

## Single cell analysis of five-prime ends with SCAFE

We used SCAFE (1.0) to analyze single nucleus 5′ paired-end chemistry libraries obtained from two DKD samples and two previously published control samples (GSE151302)[30]. Transcribed cis-regulatory elements (tCRE) were called in individual libraries using the scafe.worfklow.sc.solo function with default parameters. There was a mean of 16,912 tCRE per library (s.d. = 1855). Libraries were pooled with the scafe.workflow.sc.pool function to identify a total of 37,698 tCRE. Pooled libraries were merged into a Seurat object followed by normalization with SCTransform, batch effect correction with Harmony, dimensional reduction, and clustering. The final object contained 10,984 nuclei and cell types were annotated using snRNA-seq barcode annotations from the same samples (see snRNA-seq bioinformatics workflow above). Cell-specific tCRE and differential tCRE in diabetes were identified with the FindMarkers function with a log-fold-change threshold of 0.25. Bonferroni-adjusted $p$-values were used to determine significance at an FDR < 0.05 and significant tCRE were annotated with ChIPseeker.

## Cell culture

Human primary proximal tubular cells (human RPTEC, Lonza; CC-2553) were cultured with renal epithelial cell growth medium kit (Lonza; CC-3190). Human telomerase reverse transcriptase (hTERT)-immortalized human RPTEC (ATCC; CRL-4031) were cultured with ATCC hTERT Immortalized RPTEC Growth Kit (ATCC, ACS-4007). HEK293T cells (ATCC; CRL-3216) were cultured in Dulbecco's modified Eagle's medium (DMEM, Gibco; 11965092) supplemented with 10% fetal bovine serum (Gibco; 10437028) and antibiotics. All cultured cells were maintained in a humidified 5% $CO_2$ atmosphere at 37 °C.

## GR CUT&RUN library preparation and peak calling

CUT&RUN assay libraries for cultured cells or human kidneys were generated using the CUTANA kit (EpiCypher, 14-1048) with the manufacturer's instructions. For cultured cells, adherent cells were scraped from culture dishes and centrifuged at $500 \times g$ for 5 min. Pellets were resuspended in PBS with 1% BSA and counted. Cultured cells or nuclei obtained from a human kidney (500,000 cells or nuclei) were then mixed and incubated with Concanavalin A (ConA) conjugated paramagnetic beads. Antibodies were added to each sample (0.5 μg of rabbit glucocorticoid receptor antibody [abcam, ab225886, 1:20] or rabbit IgG negative control antibody [Epicypher, 13-0041k, 1:50]). The remaining steps were performed according to the manufacturer's instructions. Library preparation was performed using the NEBNext Ultra II DNA Library Prep Kit for Illumina (New England BioLabs, E7645S) with the manufacturer's instructions, including minor modifications indicated by CUTANA described above. CUT&RUN libraries were sequenced on a NovaSeq instrument (Illumina, 150 bp paired-end reads). Fastq files were trimmed with Trim Galore (Cutadapt [v2.8]) and aligned with Bowtie2 [v2.3.5.1] (parameters: --local --very-sensitive-local --no-unal --no-mixed --no-discordant --phred33 -I 10 -X 700) using hg38. Peak calling was performed using MACS2 [v2.2.7.1] with default parameters using samtools (1.9) and DeepTools (3.5.0). Bulk human kidney GR CUT&RUN had a mean of 24,693,221 reads (SD = 2,921,074 reads) with 62% on-target alignments and an estimated duplication rate of 0.36. Bulk human kidney IgG CUT&RUN control samples had a mean of 26,162,676 reads (SD = 1,477,128 reads) with 63% on-target alignments and an estimated duplication rate of 0.39. hTERT-RPTEC GR CUT&RUN had a mean of 21,647,282 reads (SD = 4,205,002 reads) with 75% on-target alignments and an estimated duplication rate of 0.26. hTERT-RPTEC IgG CUT&RUN control samples had a mean of 22,770,072 reads (SD = 140,623) with 66% on-target alignments and an estimated duplication rate of 0.35.

## Bulk ATAC-seq library preparation and peak calling

We suspended 50,000 cells in 50 μL of ice-cold lysis buffer with 10 mM Tris-HCl (pH 7.4), 10 mM NaCl, 3 mM MgCl₂, 1% BSA, 0.1% Tween-20 (Sigma, P7949-100ML), 0.1% NP-40 (Thermo Scientific, 28324) and 0.01% Digitonin (Thermo Scientific, BN2006)[82]. The suspension was incubated for 4 min on ice. Subsequently, 450 μL of ice-cold wash buffer (10 mM Tris-HCl [pH 7.4], 10 mM NaCl, 3 mM MgCl₂, 1% BSA, 0.1% Tween-20) was added and centrifuged at $600 \times g$ for 6 min. The pellet was resuspended in 25 μL of ATAC-seq transposition mix (12.5 μL 2 × Illumina Tagment DNA (TD) buffer; 10.5 μL nuclease-free water; 2.0 μL Tn5 transposase [Illumina, FC-121-1030]) and incubated at 37 °C for 1 h on a thermomixer. The transposed DNA was purified with MinElute PCR purification kit (QUIAGEN, 28004). DNA samples were then amplified with PCR ([72 °C; 5 min] and [98 °C; 30 s] followed by 9 cycles of [98 °C; 10 s, 63 °C; 30 s, 72 °C; 1 min] using unique 10-bp dual indexes and NEBNext High-Fidelity 2 × PCR Master Kit (M0541L). Following the first amplification, DNA size selection was performed using solid-phase reversible immobilization (SPRI) beads (AMPure XP [Beckman Coulter, A63881]) at an SPRI to DNA ratio of 0.5. The supernatant was further mixed with SPRI beads at a SPRI to DNA ratio of 1.2. The resulting supernatant was discarded, and the magnet-immobilized SPRI beads were washed twice with 80% ethanol. DNA was subsequently eluted in 20 μL of EB elution buffer (QUIAGEN, included in 28004). The size-selected DNA was amplified with an additional 9-cycle PCR. Subsequently, the amplified DNA was purified with Ampure XP (SPRI to DNA ratio of 1.7) and eluted with 25 μL of buffer EB elution buffer. The resultant ATAC-seq libraries were sequenced on a NovaSeq instrument (Illumina, 150 bp paired-end reads). Fastq files were trimmed with Trim Galore (Cutadapt [v2.8]) and aligned with Bowtie2 [v2.3.5.1] with --very-sensitive -X 2000 using hg38. PCR duplicates were removed with Picard's MarkDuplicates function. Peak calling was performed on each sample separately using MACS2

[v2.2.7.1] (--nomodel --shift −100 --extsize 200). The consensus list of accessible peaks was generated using the intersect function in bedtools. hTERT-RPTEC libraries were sequenced to a mean depth of 58,250,870 reads (SD = 13,029,194 reads) with 98% on-target alignments and an estimated duplication rate of 0.21. Primary RPTEC were sequenced to a mean depth of 67,220,749 reads (SD = 18,879,089 reads) with 98% on-target alignments and an estimated duplication rate of 0.24.

## CRISPR interference

Small guide RNA (sgRNA) targeting around the *FKBP5* TSS and intronic CRE were designed with CHOPCHOP (https://chopchop.cbu.uib.no/). These sgRNAs and two non-targeting control sgRNAs were placed following the U6 promoter in a dCas9-KRAB repression plasmid (pLV hU6-sgRNA hUbC-dCas9-KRAB-T2a-Puro, Addgene; 71236, a gift from Charles Gersbach) with golden gate assembly. The sgRNA sequences used in this study are in Supplementary Dataset 21. First, single-strand oligonucleotides (Integrated and Technology [IDT]) for sense and anti-sense sequences were annealed. Subsequently, cloning with Golden gate assembly was performed with Esp3I restriction enzyme (NEB, R0734L) and T4 DNA ligase (NEB, M0202L) on a thermal cycler repeating 37 °C for 5 min and 16 °C for 5 min for 60 cycles, followed by transformation to NEB 5-alpha Competent *E. coli* (NEB, C2987H) per manufacturer's instructions. The cloned lentiviral vectors were purified with a mini high-speed plasmid kit (IBI Scientific; IB47102). Insertion of sgRNA was checked with Sanger sequencing. For lentivirus preparation, we seeded $6.0 \times 10^5$ HEK293T cells per well on six-well tissue culture plates 16 h prior to transfection. Cells were transfected with 1.5 μg of psPAX2 (Addgene; 12260, a gift from Didier Trono), 0.15 μg of pMD2.G (Addgene; 12259, a gift from Didier Trono) and 1.5 μg of dCas9-KRAB repression plasmid per well by Lipofectamine 3000 transfection reagent (Invitrogen; L3000015) per manufacturer's instructions. Culture media were changed to DMEM supplemented with 30% FBS 24 h after transfection. Lentivirus-containing supernatants were harvested 24 h later and filtered with 0.45 μm PVDF filters (CELLTREAT; 229745). The lentivirus-containing supernatants were immediately used for lentiviral transduction. Human RPTEC were seeded at $5.0 \times 10^4$ cells per well on 6-well tissue culture plates 16 h prior to transfection. The media on human RPTEC was then changed to the fresh lentiviral supernatants supplemented with polybrene (5 μg/ml, Santa Cruz Biotechnology; sc-134220) and cultured for 24 h. Subsequently, RPTEC cells were cultured in renal epithelial cell growth medium and puromycin (3 μg/ml, invivogen; ant-pr-1) for 72 h.

## Quantitative PCR

RNA from human RPTECs was extracted with the TRIZOL and Direct-zol MicroPrep Plus Kit (Zymo) following the manufacturer's instructions. Extracted RNA (1-2 μg) was used for reverse transcription to generate cDNA libraries with the High-Capacity cDNA Reverse Transcription Kit (Life Technologies). Quantitative PCR was performed in the BioRad CFX96 Real-Time System using iTaq Universal SYBR Green Supermix (Bio-Rad). Expression levels were normalized to *GAPDH*, and data were analyzed using the 2-ΔΔCt method. Quantitative PCR data are presented as mean±s.d. and compared between groups with one-way ANOVA and a post-hoc Dunnett's adjustment for multiple comparisons. A *p*-value < 0.05 was considered statistically significant. Primer sequences are provided in supplementary materials (Supplementary Dataset 21).

## Bulk RNA-seq analysis of previously published human DKD

Raw fastq files were downloaded from GSE142025 to include 9 control, 6 early DKD, and 22 advanced DKD donors[43]. Transcript abundance was quantified with Salmon (1.8.0) using Ensembl (release-99) and count matrices were imported to DESeq2 (1.32.0) with tximport (v1.16.1). Differentially expressed genes were identified using the

DESeq function with default parameters for early DKD vs. Control and advanced DKD vs. Control (Supplementary Dataset 18). Significance was determined using a Benjamini-Hochberg adjusted $p$-value.

### Comparison of Cell-specific DAR and GR binding sites to differentially methylated regions in CKD or DKD

Differentially methylated regions (DMR) were downloaded from publicly available databases comparing control kidney samples to CKD or DKD[44–48]. DMR were lifted over to hg38 coordinates using the UCSC LiftOver utility and compiled into an aggregated list (Supplementary Dataset 19; sheet = "ALL_DAR"). DMR were flanked by a 1 kb window and intersected with cell-specific DAR and GR CUT&RUN binding sites using GenomicRanges in R.

### Partitioned heritability of ATAC peaks

Cell-specific ATAC peaks and cell-specific DAR in diabetes were identified with the Seurat FindMarkers function, sorted by $p$-value, and filtered for peaks with an average log-fold-change greater than zero. All peaks that met the adjusted $p$-value threshold were used to generate a cell-specific bed file. In the event a cell type did not have at least 2000 peaks that met the adjusted $p$-value threshold, the top 2000 peaks with the lowest $p$-value were used to create the bed file. If a cell type did not have 2000 cell-specific peaks, all available peaks were used to create the bed file. For the allele-specific analysis, bed files were generated using peaks that met the adjusted binomial threshold for allelic bias of reference vs. alternate allele ($N = 5593$, $p$adj < 0.05) or the unadjusted $p$-value threshold for the base model ($N = 7512$, $p$val < 0.05), model 2 ($N = 7557$, $p$val < 0.05), and model 3 ($N = 7353$, $p$val < 0.05). These thresholds were used to keep the number of peaks in each annotation roughly equivalent. Bed files were lifted over to hg19 to create annotations for autosomal chromosomes with a 1000 genomes phase 3 reference and the make_annot.py function in ldsc (1.0) using a 100 kb window[54]. Linkage disequilibrium scores were computed from custom annotations with the ldsc.py function using default parameters. GWAS summary statistics for eGFR, CKD, micro-albuminuria, and urinary sodium excretion were downloaded from publicly available databases and formatted for ldsc using munge_sumstats.py[51–53]. Partitioned heritability for each GWAS trait was estimated using the 1000 G phase 3 reference and ldsc cell-type-specific workflow with default parameters, including baseline v1.2 annotations after controlling for all kidney ATAC peaks in the dataset. $P$-values were adjusted for multiple comparisons using Benjamini-Hochberg and significance was determined at $p$adj < 0.05.

### Allele-specific modeling with SALSA

Coordinate-sorted bam files generated by cellranger (snRNA-seq) or cellranger-atac (snATAC-seq) were genotyped with SALSA using GATK (4.2.0.0) best practices for germline short variant discovery[57]. For snRNA-seq, reads containing Ns in their cigar string (e.g. spanning splice junctions in snRNA-seq data) were split using SplitNCigarReads. For snRNA-seq and snATAC-seq, base recalibration was performed with BaseRecalibrator using hg38 GATK bundle resources, including dbsnp (v138), 1000 G phase I indels, 1000 G phase I high-confidence SNV, and Mills and 1000 G gold standard indels. Recalibration was applied with ApplyBQSR to create analysis-ready bam files. Variants were identified from analysis-ready bam files with HaplotypeCaller and genotypes were called from GVCFs using GenotypeGVCFs with default parameters. snRNA-seq variants were hard-filtered by Fisher strand bias (FS > 30), quality by depth score (QD < 2), cluster size[3], and cluster-window size (35 bp). snATAC-seq were filtered using CNNScoreVariants followed by FilterVariantTranches (--snp-tranche 99.95 indel-tranche 99.4). snRNA-seq libraries had a mean total of 98,255 (s.d. = 64,847) SNVs and indels and snATAC-seq libraries had a mean total of 2,283,904 (s.d. = 534,165) SNVs and indels.

Genotypes from snRNA-seq and snATAC-seq were combined and phased with shapeit4.2 using the 1000 Genomes phased reference for biallelic SNV and indels on GRCh38[57]. There was a mean of 1,917,939 (s.d. = 363,223) phased SNV and indels for each library, which were used to perform variant-aware realignment with WASP (0.3.4) using bwa (0.7.17) or STAR (2.5.1b) and filtered with bcftools (1.9)[59]. WASP-aligned bam files were divided into single cell bam files by extracting proximal-tubule-specific barcodes using the CB tag. GATK ASEReadCounter was used to generate single cell allele-specific counts from single cell bam files using phased heterozygous SNV (mean = 722,091, s.d. = 199,756).

Pseudo-multiomic cells were created by performing label transfer from the aggregated snRNA-seq to snATAC-seq dataset to generate an imputed RNA estimate for each snATAC-seq cell. One or more gene targets for each ATAC peak were identified using the LinkPeaks function. ATAC peaks with heterozygous SNV were filtered for an aggregate total fragment count >20, total reference allele count >5, and total alternate allele count >5. The ratio of aggregated reference counts to alternate counts within ATAC peaks containing heterozygous SNV was compared with a binomial test. A generalized linear mixed effect model with a logit link function was implemented with the lme4 package[83]. In the base model, the dependent variable was coded as the presence of an alternate allele within an ATAC peak and the continuous predictor variable was the imputed RNA estimate normalized to an interval from 0 to 100. A mixed effect per sample was added to control for pseudo-replication bias. In model 2, an additional fixed effect for diabetes was added. In model 3, additional fixed effects for diabetes and an interaction term between diabetes and imputed RNA expression were added. The significance of each peak-gene combination was evaluated using a Wald test obtained from the glmer function. In the supplementary materials, all fixed-effect coefficient estimates for each peak-gene combination are included with 95% confidence intervals, Wald $p$-values, and standard deviation estimates of random effects (Supplementary Dataset 20). Peak-gene combinations meeting the nominal $p$-value threshold for expression ($p < 0.05$) were annotated with ChIPseeker and the corresponding effect size in log-odds is visualized in relation to the nearest TSS. These same peaks were also used to create annotations and calculate linkage disequilibrium scores with ldsc for partitioned heritability of eGFR as previously described.

### Reporting summary

Further information on research design is available in the Nature Research Reporting Summary linked to this article.

## Data availability

All of the data for this manuscript have been made publicly available. Raw sequencing data for snATAC-seq ($N = 1$ control, $N = 7$ DKD) and snRNA-seq ($N = 1$ control, $N = 2$ DKD) is deposited in GEO under accession number "GSE195460". Previously published raw sequencing data for snRNA-seq ($N = 5$ control, $N = 3$ DKD) and snATAC-seq ($N = 5$ control) are available in "GSE151302" and "GSE131882". Processed count matrices for all snRNA-seq ($N = 11$) and snATAC-seq ($N = 13$) libraries used in this study are also provided in "GSE195460". Sequencing data for CUT&RUN from bulk kidney cortex and primary RPTEC are deposited under accession number "GSE195443". Sequencing data for Omni-ATAC from hTERT-RPTEC and primary RPTEC are also deposited under accession number "GSE195443". Gene expression and chromatin accessibility for each cell type can be viewed on our interactive website; "Kidney Interactive Transcriptomics [http://humphreyslab.com/SingleCell]" (dataset: Wilson and Muto et al). All other relevant data supporting the key findings of this study are available within the article and its Supplementary Information files or from the corresponding author upon reasonable request. Source data are provided with this paper.

 

## Code availability

SALSA is available on GitHub (https://github.com/p4rkerw/SALSA)[84]. All of the analysis code used to generate data in this manuscript is available on GitHub (https://github.com/p4rkerw/Wilson_Muto_NComm_2022)[85].

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

## Acknowledgements

These studies were supported by a pilot grant from the Washington University Diabetes Research Center (NIH DK20579) and an NIH K08 Career Development Award (NIH DK126847) to P.C.W. These studies were also supported by an NIH R01 (DK103740) and a Chan Zuckerberg Initiative seed network grant (CZF2019-002430) to B.D.H. Additional support was provided by the International Research Fund for Subsidy of Kyushu University School of Medicine Alumni, the Japan Society for the Promotion of Science (JSPS) Postdoctoral Fellowships for Research Abroad, and the Osamu Hayaishi Memorial Scholarship for Study Abroad to Y.M.

## Author contributions

P.C.W. and Y.M. contributed equally to this study. P.C.W., Y.M., and B.D.H. planned the study. S.S.W. and A.K. collected the kidney samples. Y.M. prepared the snRNA-seq and snATAC-seq libraries. P.C.W., Y.M., and B.D.H. analyzed and interpreted snRNA-seq and snATAC-seq data. Y.M. performed other experiments. H.W. generated the public database for data. P.C.W., Y.M., and B.D.H. wrote the manuscript. B.D.H. coordinated and oversaw the study. All authors discussed the results and commented on the manuscript.

## Competing interests

The authors declare no competing interests.
