## [Peer Review File · Nature Communications]

REVIEWER COMMENTS

Reviewer #1 (Remarks to the Author):

Wilson et al represent a joint human kidney snATAC-seq and snRNA-seq of control and DKD samples and highlight alteration in GR in kidneys of diabetic subjects. They also provide CUT&RUN as well as CRISPRi model to validate the findings. While focusing on the roles of GR in the DKD is novel, a fundamental problem is the lack of robust evidence that why the authors picked GR. They focus on FKBP5 within several target regions of glucocorticoids and it is not obvious why this region was picked. Indeed, it seems that the authors arbitrarily without too much supporting evidence wanted to focus on GR and FKBP5. Another key issue the paper does not have a linear story and each part somehow separate from other making it difficult to follow. Also, the data here is from a previously published snRNA-seq and snATAC-seq with only a few added samples taking away from the novelty of the work.

Overall my comments are as follows:

Major comments:

-Introduction: It is too long with un-necessary details as well as disorganized flow. For example, in the first and last paragraph the authors mentioned their aim and ultimate findings. It must be re-written.

-The definition of DKD in this paper is questionable. The eGFR in the DKD samples were around 60 without obvious kidney failure. The only way to confirm the diagnosis of DKD is GBM thickness measurements via EM which is missing from the presented work. The presented pathological findings could be non-specific, no KW nodules presented. Also, only 2 of the diabetic subjects had proteinuria raising the question whether the samples were real DKD. It seems it would be better to describe these samples as diabetic but NOT DKD.

The authors should show at least a minimum of quality control plots regarding their snATAC and snRNA-seq datasets to demonstrate distribution of sequencing depth, nUMI/cell, nfeatures/cell, adequate doublet detection, independence of ambient RNA effects, and coherence of integration of batches/samples.

In the methods section, the snATAC-seq and snRNA-seq analysis parts are rather superficial. In order to facilitate reproducibility of their results, in addition to sharing the entire code used to produce their datasets and plots, the manuscript should mention in more detail individual function parameters, e.g. k, resolution, etc.

The raw and processed data are not accessible to the reviewer, the GSE accession numbers are crossed out.

- Fig 1- A small blue cluster which was attached to the PT-VCAM-1 which is the same as the PEC cells. Do you mean that you had two PEC clusters? What is that small cluster? It seems that your PEC cluster based on your dotplot (supplementary fig 2A) is not a pure PEC and it is contaminated and express

several markers (maybe doublets). Or the separate cluster that you annotated as PEC is not a real PEC cluster because always PEC comes close to PT-VCAM-1. This annotation should be checked carefully.

-Fig 1: For PST cluster you used SLC5A1 which expressed in other clusters and not specific. Is it real PST? You should show more specific markers (SLC7A13, SLC5A8) if it is real PST.

-Your statistical cut-off is not stringent. You selected the log2FC of 0.1 for DAR as significant which is different with your cut-off with DEG. It does not make sense. You should increase your DAR cut-off as 0.25 (likely most of your DARs will be disappear (according to figure 1B)).

-Are the assumptions (e.g. normality) for a t test comparing 5 vs. 6 PT_VCAM1 fractions really satisfied? I suspect a non-parametrical test would be more appropriate.

-Fig 1E. I do not really see any difference between INSR chromatin accessibility between control and DKD. What was the log2FC? The thing that I can see is that in the promotor region the accessibility of INSR in DKD was higher (according to your coverage plot). Please clarify this.

-Fig 1F. How did you measure 20% less? Please clarify it.

-There are several mistakes of the interpretation of FoldChange. For example you mentioned "Similar to the diabetic proximal tubule, there was decreased expression of INSR (fold-change = 0.76, padj = 1.5×10^{-17})" when fold-change is positive it means you have increased expression; in that case it was 1.76 times more expression. Also,:

"There was also decreased expression of HSD11B2 (fold-change = 0.71, padj = 5.3×10^{-33}), which is the enzyme that catalyzes the conversion of cortisol to the inactive metabolite cortisone to protect nonselective activation of MR"

"a marked reduction in FKBP5 (fold-change = 0.45, padj = 4.9×10^{-117})"

This seems to an obvious and significant mistake and raising issues around the conclusion of the paper.

-Fig 2A: You had two PC clusters which do not make sense. PC cluster always close to the DCT and CNT. However, you showed a cluster attach TAL and called it as PC. On the other hand, in the dotplot AQP2 expression was found only in 50% of that cluster which is not possible as most of the PC express AQP2. Also, if both of them PC why did you use two different colors? It seems that the cluster attached to TAL is macula densa. You should consider re-annotating the data.

-Fig 2E: Again I cannot see any obvious difference in the promoter of PCK1; even the accessibility is higher in the control.

-I am unsure about the additional value of tCRE analysis, as from the text it is not quite evident what orthogonal information it really provides. The argument that PT_VCAM1-specific tCRE shows enhanced transcription of VCAM_1 is also not surprising. I am lost with this part

-The comparison of PT_VCAM1 fractions between DKD and Control samples gives a p of 0.14, which indicates no statistically significant difference, the authors should not claim a "trend". The more interesting question would be if the authors can compare the fraction of these cells to non-diabetic CKD samples. What is the authors' take on specificity of the PT_VCAM1 population for DKD development, progression?

-No QC reported for CUT&RUN data. Please provide the QC.

-Fig5E- the decreased accessibility of FKBP5 is not obvious in diabetic samples. Even in the promoter I see higher accessibility in diabetic samples.

-The authors' rigidity of analytical approach seems questionable at times: E.g., in the scRNAseq results part, they talk about certain glucose pathway genes differentially expressed after showing enrichment for glucose and glucocorticoid pathways. Important to note that the genes the authors mention did not end up among the top DEGs in the first place, as the log fold changes are miniscule. This reviewer cannot help but get the impression that the authors – probably for the sake of presenting a coherent storyline – sometimes sacrifice objectivity of bottom-up data analysis in favor of a top-down approach, zooming in a lot on tiny fold changes between a single cell type of interest (in this case PT) when analyzing DEGs between DKD and control samples. A log₂ FC of 0.19 in the case of NR3C1 seems somewhat underwhelming. Rather, it would be of interest what the log₂ fold changes for this gene are in other tubular segments (PT_VCAM1, LOH, DCT, collecting duct, etc.) and compare them head-to-head. So far, I am not fully convinced that these changes are relevant to justify highlighting them (just because they fit the ATAC analysis, for that matter) or that they are even cell type-specific. Along those lines, while it is true that the PT is important for glucose reabsorption and metabolism, it needs to be pointed out that – especially given the meager log₂FC for, e.g., SGLT2 or PCK1, most other (also tubular) cells in the kidney also feed on and produce and metabolize glucose, for which they import glucose via various transporters (GLUTs, SGLT1, etc.) and process with enzymes etc. A comprehensive analysis of glucose transport, production, and metabolism with focus on effect size and cell type specificity is warranted to exclude compensatory effects in other (tubular) cell types than PT that might even be larger than the presented effects in PT.

Most importantly there is a lack of a clear message in here.

Minor comments

-Why did you use 17 cycles for cDNA preparation of your single cell libraries? According to the protocol the maximum should be 13. It can seriously alter the validity of the results.

-The demultiplexing protocols for your snATAC-seq and snRNA-seq libraries should be mentioned.

-Please provide the data QC both for the snATAC-seq and the snRNA-seq datasets.

What statistical test was used for comparison? “There was a trend towards greater proportion of PT_VCAM1 in DKD samples compared to healthy controls (0.09 vs. 0.25, $p = 0.14$).”

-How did you compare the fractions with student's t test in this part? “There was a trend toward greater proportion of PT_VCAM1 in DKD compared to healthy controls (mean proportion 0.06 vs. 0.02, 241 Student's t-Test $p=0.051$).” You could use chi-square.

-Please provide the UMAPs of both your dataset in terms of original clustering as well as representation of samples to show the batch effect.

-Ambient RNA is very important in snRNA-seq that could affect the results. Was the data corrected for ambient RNA if not why?

Fig. 2B bears almost no additional information to the text and could be omitted. Maybe certain genes could be highlighted to make it useful in some way and help focus?

Fig. 2F: hard to see any changes in the violin plots. Dot plots might be better suited.

It's transposase-accessible chromatin, not transpose-accessible chromatin

Reviewer #2 (Remarks to the Author):

To identify the altered signaling pathways and transcription factors associated with diabetic kidney disease (DKD), the authors generated single nucleus RNA-sequencing (snRNA-seq) and assay for transposase accessible chromatin sequencing (snATAC-seq) on human kidney cortex from donors with and without DKD. The authors identified increased proportion of inflammatory VCAM1+ injured proximal tubule cells (PT_VCAM1) in DKD. snATAC seq data identified the cell-specific differentially accessible regions (DAR) in nephron in DKD. DAR enriched for glucocorticoid receptor (GR) motifs. The authors found that changes in chromatin accessibility are associated with decreased insulin receptor expression, increased gluconeogenesis, and decreased expression of the GR cytosolic chaperone, FKBP5, in the diabetic proximal tubule. Cleavage under targets and release using nuclease (CUT&RUN) of GR in control cortex tissue and renal proximal tubule epithelial cells (RPTEC) showed co-localization of DAR and GR binding sites.

Moreover, the authors demonstrated that CRISPRi silencing of GR response elements (GRE) in the FKBP5 gene body reduced expression of FKBP5 (a negative regulator of GR) in RPTEC. In addition, the authors showed that genetic background might regulate chromatin accessibility and DKD progression. Moreover, the authors developed open-source tools for single-cell allele-specific analysis (SALSA) to model the effect of genetic background on gene expression.

Overall, the manuscript is well written, and the conclusions are derived from several state-of-art techniques and analyses and largely supported by the results. This is one of the first studies to perform integrative multimodal single nucleus sequencing in DKD kidney samples. The findings provide a better understanding of the single-cell landscape and genetic heritability in DKD. They are also helpful to other researchers to identify the cell-specific changes in the gene of interest in their kidney datasets. I have outlined a number of aspects to be addressed to improve the paper.

General Comments:

1. Abstract: Can be more informative to highlight the major objectives and findings of the study and the key conclusions. The current Abstract provides technical details without providing the rationale or broad conclusions and clinical implications.
2. The authors recently (Nat Commun 2021, Muto et al) used healthy adult human kidneys to perform sn-RNA-seq and sn_ATAC-seq. Presumably, some of those datasets are also included in the current study (ie the control samples). In this previous study, they also found a subpopulation of proximal tubule that had increased expression of VCAM1 (PT_VCAM1) as well as HAVCR1 (kidney injury molecule-1). So it appears these marker of tubular injury are enriched even in “healthy” kidney PCTs. Can the authors comment on this? Given the relatively small n’s, and without correction for various variables (including HbA1c, blood pressure, age, etc) could some of the changes be also attributed to one or more of these confounders?
3. In Figure 3B – The authors noticed a significant proportion of tCRE in intronic (Figure 3B, 11847/37,698, 302 31%) and intergenic regions (Figure 3B, 1367/37,698, 3%) which they speculate may be representative of enhancer RNA (eRNA). In general, eRNAs are not easily captured by single cell sequencing technologies due to low or cell specific expression or rapid turnover. Please comment.
4. All Supplemental tables are missing from the manuscript (Will these be provided on request?).
5. Line no. 159-167: Provide the data describing in these lines, which might be helpful for readers.
6. In the Supplement Figure, provide the data for proportions of the cell's numbers identified in control and DKD samples for snATAC-sq and snRNA-seq.
7. In the Supplementary Fig, provide the data for ATP1B1 and KCNE1B, like figure 1E.
8. A similar figure like figure 2E can be generated and included in the supplementary figure for ALDOB, FBP1, and G6PC.
9. The authors showed the RNA expression of GR and FKBP5 in the manuscript. Is the protein expression of GR and FKBP5 altered in control versus DKD samples? This could be tested by western blot/ immunohistochemistry analysis.
10. In lines 675-678, the authors hypothesize that DNA hypermethylation might be changed (hypermethylated) near FKBP5. Authors can provide DNA methylation data at the differential snATAC peaks in FKBP5. Authors can also check if publicly available data sets on DNA methylation in diabetes/ cardiovascular risk overlap with the differentially accessible regions (DAR) from snATAC-seq/GR peaks.
11. Authors can provide the list of anti-inflammatory effect genes as hypothesized in Figure 8.

Reviewer #3 (Remarks to the Author):

Summary: The authors present an analysis of alterations in single-nucleus chromatin availability and transcriptional signature in DKD vs. control samples. Key findings include a trend towards an increased proportion of VCAM1+ proximal tubule cells in DKD samples, cell-type specific ATAC peaks showing enrichment for heritability of kidney-related traits, enrichment of glucocorticoid and mineralocorticoid receptor motifs in genomic regions differentially accessible in DKD, and association of SNVs in a subset of ATAC peaks with alterations in expression of the the target genes. The manuscript represents a significant advance on previous work, providing to my knowledge the first single-nucleus ATAC-seq profiling of diabetic kidney disease samples, and is generally thoughtfully written, interesting, and clear.

Comments:

It would be interesting to know for several of the analyses how the reported trends are distributed across patients and between control vs. DKD patients. For example:

- How many cells of each type come from each patient for snRNA-seq and ATAC-seq datasets?
- What is the distribution of PT_VCAM1 proportion across DKD and control patients for the snATAC-seq and snRNA-seq datasets?
- What is the distribution of average INSR expression in PT across DKD and control patients?
- For heterozygous SNVs that appear across multiple patients, how consistent are the estimated effects on gene expression across individuals?

Why is the number of GR binding sites specific to hTERT-RPTEC compared to IgG-stimulated control samples (22,539) so much larger than the number of GR binding sites identified in bulk kidney cortex from a healthy donor (4,362)? How do the location of the binding sites compare between hTERT-RPTEC and healthy donor kidney?

Line 514-515: SALSA seems to be similar or perhaps extended from a method previously described by the authors in (Muto et al. Nat Commun. 2021 Apr 13;12(1):2190) and indeed the github page for SALSA suggests the previous manuscript as the citation. The original citation should be provided on lines 514-515 and the text should explain what has changed in the method from the previous work. (Is this a repackaging of the previous code to make it more usable, or has the method substantially changed from the previous approach)?

Minor comments:

Abstract line 37-38: The increased proportion of VCAM1+ cells does not pass a significance threshold in either snRNA-seq or sn-ATAC seq, so consider reporting it in the abstract as “a trend towards an increased proportion of VCAM1+ injured proximal tubule cells” as in the text.

Line 151: The control patients cannot really be described as “healthy” as these samples are from tumor nephrectomy and deceased organ donors. Perhaps describe these patients as just “control patients.”

Line 178: It looks like it is Supplemental Figure 2 (not Supplemental Figure 1) that shows identification of cell types “based on increased chromatin accessibility within gene body and promoter regions of lineage specific markers.”

Line 184-5: Are these means of the cell type proportion across samples?

Figure 1E caption: This is a very modest fold change (0.92; log-fold change -0.11). Presumably the significant p-value is possible because of the large number of cells analyzed. Is the direction of the fold change seen consistently across pairwise comparisons between the DKD patients and the healthy patients?

Figure 1F caption: The text appears to be truncated--the caption ends with “(fold-”. Presumably the fold-change and p-value would be provided in the rest of the caption.

Lines 306, 309: The wrong supplemental table appears to be referenced here; Supplemental Table 7 has the comparison between PT and PT_VCAM markers, not the cell type specific tCRE.

Line 338: Was this sample obtained from one of the 6 control donors from the snRNA-seq analysis? Consider providing the anonymized id (e.g “Control1”) so that the reader can cross-reference to the metadata in Supp Table 1 (or provide anonymized metadata for this donor).

Figure 4A legend: What does “distance between the peak and its two neighboring regions” mean? Is this minimal distance to a genic region?

Figure 6 appears to be lower-resolution than other figures in the paper.

Lines 551, 560, 563, 572, 623, 625: should these be “nominally significant”/ “nominal p-value threshold” since these peak-gene combination counts appear to be without multiple hypothesis testing correction?

Line 557: “this translates to the typical ATAC peak being 1.10 times more likely to contain an alternate allele in the base model:” This seems like a very modest increase in the probability of containing an alternate allele, and I am surprised that such a small allele bias would lead to a 10% change in gene expression. Is the text correct as stated?

Figure 7A, B: Although no cell types pass the significance threshold for microalbuminuria, the color of all of the bars is different from the color of the the non-significant bars in the other plots.

Line 628: I don't think that the acronym for generalized linear mixed model (GLMM) is defined anywhere in the manuscript.

I would suggest including in the supplemental materials an index of the supplemental tables with a brief description of the content of each table.

Reviewer #1 (Remarks to the Author):

Wilson et al represent a joint human kidney snATAC-seq and snRNA-seq of control and DKD samples and highlight alteration in GR in kidneys of diabetic subjects. They also provide CUT&RUN as well as CRISPRi model to validate the findings. While focusing on the roles of GR in the DKD is novel, a fundamental problem is the lack of robust evidence that why the authors picked GR. They focus on FKBP5 within several target regions of glucocorticoids and it is not obvious why this region was picked. Indeed, it seems that the authors arbitrarily without too much supporting evidence wanted to focus on GR and FKBP5. Another key issue the paper does not have a linear story and each part somehow separate from other making it difficult to follow. Also, the data here is from a previously published snRNA-seq and snATAC-seq with only a few added samples taking away from the novelty of the work.

Overall my comments are as follows:

Major comments:

-Introduction: It is too long with un-necessary details as well as disorganized flow. For example, in the first and last paragraph the authors mentioned their aim and ultimate findings. It must be re-written.

We have rewritten and reduced the length of the introduction.

-The definition of DKD in this paper is questionable. The eGFR in the DKD samples were around 60 without obvious kidney failure. The only way to confirm the diagnosis of DKD is GBM thickness measurements via EM which is missing from the presented work. The presented pathological findings could be non-specific, no KW nodules presented. Also, only 2 of the diabetic subjects had proteinuria raising the question whether the samples were real DKD. It seems it would be better to describe these samples as diabetic but NOT DKD.

We appreciate the reviewers' comments, but would like to note that all of the samples analyzed in this study were independently reviewed by two renal pathologists using the renal pathology society classification for diabetic nephropathy (PMID: 20167701). GBM thickness is only one component of the RPS classification and each of the examined samples had evidence of both GBM thickening, mesangial expansion, and mesangial hypercellularity. A subset of samples had more advanced disease with nodular mesangial expansion and mild to moderate interstitial fibrosis. In this manner, each of the biopsies was classified as either RPS class II or III. EM measurements are only needed to diagnose isolated GBM thickening (RPS class I). Reduced eGFR and/or proteinuria are not components of the RPS classification scheme, but the presence of proteinuria in a subset of patients is not unusual for a highly heterogeneous disease. Histologic findings often precede clinical manifestations (PMID: 31167184). We have included representative histologic images and all of the pathology and laboratory metadata is available in the supplementary material (Supplemental Table 1).

The authors should show at least a minimum of quality control plots regarding their snATAC and snRNA-seq datasets to demonstrate distribution of sequencing depth, nUMI/cell, nfeatures/cell, adequate doublet detection, independence of ambient RNA effects, and coherence of integration of batches/samples.

We have generated the requested QC plots to include the distribution by sequencing depth, number of features per cell, doublet detection results, and batch effect correction. These new plots are now in the supplementary materials (Supplemental Tables 10 and 11).

We used SoupX [PMID: 33367645] to estimate ambient RNA contamination in the aggregated snRNA-seq object and obtained a global rho of 0.04. This estimate represents the proportion of all barcodes in the dataset affected by ambient RNA contamination and is a low proportion. When we limited the SoupX analysis to only annotated barcodes that met our quality control metrics, the proportion of cells affected by ambient RNA contamination remained at 0.04. We used the adjusted count matrix from SoupX to recluster our data using the same parameters as the unadjusted dataset and did not observe any significant changes in cell type annotation (see UMAP below).

We have deposited the additional code for this analysis in the GitHub repository at the following location [Wilson_Muto_NComm_2022/blob/main/snRNA_prep/step3_soupx.R](https://github.com/Wilson_Muto_NComm_2022/blob/main/snRNA_prep/step3_soupx.R).

In the methods section, the snATAC-seq and snRNA-seq analysis parts are rather superficial. In order to facilitate reproducibility of their results, in addition to sharing the entire code used to produce their datasets and plots, the manuscript should mention in more detail individual function parameters, e.g. k, resolution, etc.

We have added more detail to the snATAC-seq and snRNA-seq analysis methods section. We have also shared all of the code through our GitHub repository. Our entire pipeline has been run in publicly-available docker containers to ensure reproducibility.

The raw and processed data are not accessible to the reviewer, the GSE accession numbers are crossed out.

The raw and processed data are available in GEO and are only accessible when using a reviewer token. The reviewer tokens for accessing the data is in the Data Availability section in the manuscript.

- Fig 1- A small blue cluster which was attached to the PT-VCAM-1 which is the same as the PEC cells. Do you mean that you had two PEC clusters? What is that small cluster? It seems that your PEC cluster based on your dotplot (supplementary fig 2A) is not a pure PEC and it is contaminated and express several markers (maybe doublets). Or the separate cluster that you annotated as PEC is not a real PEC cluster because always PEC comes close to PT-VCAM-1. This annotation should be checked carefully.

We agree with the reviewer that annotation of this cluster can be improved. On close inspection, the cluster labeled “PEC” adjacent to PT_VCAM1 (cluster #18) has increased activity for *PROM1* (CD133), very low activity for *CFH*, and limited activity for *HAVCR1* and *SLC5A1*. We had initially annotated this cluster using label transfer from the snRNA-seq dataset where approximately half of the cells had a predicted id of PEC and the other half were PT. After review, we believe that the cluster represents an injured population of PT (and not PEC as we had previously indicated). It does not appear to express any other cell-specific markers in the kidney that would suggest the presence of doublets. We used FindMarkers to detect cluster-specific markers and this cluster has cell-specific gene activity for *PROM1* (*PROM1*^{high} *VCAM1*-) to an even greater extent than PT_VCAM1 (*PROM1*^{low} *VCAM1*+). We have updated the figure and annotated this cluster as PT_PROM1 and believe that it represents a population of CD133+VCAM1- cells that we previously identified by immunofluorescence in our earlier work [Muto et al. PMC8044133]. We think it’s likely that there are multiple subpopulations of injured proximal tubules with unique chromatin accessibility and protein expression profiles and have added some detail to the manuscript to discuss these findings. The cluster labeled “PEC” (cluster #15 below) that is non-adjacent to PT_VCAM1 has high gene activity for *CFH* and is correctly annotated. We appreciate this comment.

-Fig 1: For PST cluster you used SLC5A1 which expressed in other clusters and not specific. Is it real PST? You should show more specific markers (SLC7A13, SLC5A8) if it is real PST.

The PST has the expected expression pattern and is SLC5A1^{high} SLC5A2^{low} SLC7A13^{high} SLC5A8^{low} compared to the PCT (see dotplot below). SLC5A1 is expressed predominantly in PST.

-Your statistical cut-off is not stringent. You selected the log2FC of 0.1 for DAR as significant which is different with your cut-off with DEG. It does not make sense. You should increase your DAR cut-off as 0.25 (likely most of your DARs will be disappear (according to figure 1B)).

The 0.1 effect size threshold for the snATAC-seq analysis has no relationship to statistical significance. Statistical significance is determined by the adjusted p-value threshold and there are many additional peaks in our analysis that reach the adjusted p-value threshold that are below our designated effect size threshold. For the snATAC-seq analysis, we selected a log2FC effect size threshold of 0.1 because the dynamic range of chromatin accessibility is much less than that of gene expression. There is no established effect size threshold for measuring changes in chromatin accessibility, however, we can estimate an appropriate effect size threshold by looking at the magnitude of cell-specific peaks for a given cell type. If we take the top 100 most accessible peaks in PCT from the snATAC-seq dataset (relative to other cell types), the median log2FC is 0.42 (fold-change = 1.33). If we take the top 100 most upregulated genes in PT from the snRNA-seq dataset (relative to other cell types), the median log2FC is 2.2 (fold-change = 4.5). This suggests that the snRNA-seq markers have a much greater dynamic range compared to snATAC-seq and that the effect size threshold of snATAC-seq should be set much lower than snRNA-seq. In any case, we have provided all of the results of our marker analysis in the supplementary data so researchers can use any threshold they like.

-Are the assumptions (e.g. normality) for a t test comparing 5 vs. 6 PT_VCAM1 fractions really satisfied? I suspect a non-parametrical test would be more appropriate.

We agree with the reviewer that a non-parametric test may be more appropriate here. Therefore we now use a Wilcoxon rank sum test to compare the proportion of PT_VCAM1 relative to the total number of cells for 6 control vs. 7 DKD snATAC-seq samples and obtained $p=0.004$. We used the same approach to compare the proportions between 6 control and 5 DKD snRNA-seq samples and obtained $p=0.03$. We have updated the manuscript to use the non-parametric test.

-Fig 1E. I do not really see any difference between INSR chromatin accessibility between control and DKD. What was the log₂FC? The thing that I can see is that in the promoter region the accessibility of INSR in DKD was higher (according to your coverage plot). Please clarify this.

The control ATAC peak is above the second grid line and the DKD ATAC peak is below the second grid line as indicated by the orange arrow. The log₂FC for DKD vs. control is -0.11 (corresponding to decreased chromatin accessibility). The INSR promoter region in DKD does not appear to be higher than control samples to our eye, and there are no statistically-significant regions that change accessibility other than the region we indicated. We would like to note that it's difficult to draw conclusions from the coverage plots because they are downsampled before plotting.

-Fig 1F. How did you measure 20% less? Please clarify it.

The 20% reduction in INSR expression was obtained by comparing control PT to diabetic PT using the snRNA-seq dataset and the FindMarkers function with the default Wilcoxon rank sum test. We converted a log₂FC of -0.34 to fold-change by taking $2^{\log_2\text{FC}}$ and obtained a value of 0.79, which corresponds to an approximate 20% decrease.

-There are several mistakes of the interpretation of FoldChange. For example you mentioned "Similar to the diabetic proximal tubule, there was decreased expression of INSR (fold-change = 0.76, padj = 1.5×10^{-17})" when fold-change is positive it means you have increased expression; in that case it was 1.76 times more expression. Also, "There was also decreased expression of HSD11B2 (fold-change = 0.71, padj = 5.3×10^{-33}), which is the enzyme that catalyzes the conversion of cortisol to the inactive metabolite cortisone to protect nonselective activation of MR" "a marked reduction in FKBP5 (fold-change = 0.45, padj = 4.9×10^{-117})" This seems to an obvious and significant mistake and raising issues around the conclusion of the paper.

We appreciate the reviewer comments, but believe that you may be confusing fold-change with log-fold-change. Fold-change is the ratio between DKD and control samples and is always positive by definition. For example, a fold-change of 2 indicates that DKD had twice as much expression as control samples and a fold-change of 0.5 means that DKD had half as much expression as control samples. If we take the logarithm of the fold-change (log-fold-change) then we get a log₂FC of 1 for a fold-change of 2 and log₂FC of -1 for a fold-change of 0.5. In the text, we used fold-change because it's easier for readers to interpret. For example, a fold-change of 0.76 corresponds to a 24% reduction and a log-fold-change of -0.39. We have double-checked the manuscript and there are no mistakes in the interpretation of fold-change.

-Fig 2A: You had two PC clusters which do not make sense. PC cluster always close to the DCT and CNT. However, you showed a cluster attach TAL and called it as PC. On the other hand, in the dotplot AQP2 expression was found only in 50% of that cluster which is not possible as most of the PC express AQP2. Also, if both of them PC why did you use two different colors? It seems that the cluster attached to TAL is macula densa. You should consider re-annotating the data.

Thank you for your close attention to detail. The “PC” cluster adjacent to TAL1 in Fig2A was mislabeled. The label has been changed to ATL as was originally intended.

-Fig 2E: Again I cannot see any obvious difference in the promoter of PCK1; even the accessibility is higher in the control.

We indicated in the figure that DKD shows decreased accessibility relative to control (ie. control is higher) as you pointed out. Part of the disconnect may stem from the fact that increased chromatin accessibility does not always translate to increased gene expression. We believe this is an example where decreased chromatin accessibility is associated with increased gene expression in the proximal tubule. In particular, we would like to note that decreased chromatin accessibility in DKD corresponds to all of the fragments that map to the peak region indicated by the orange bars along the bottom of the plot. It is the aggregate sum of fragments across that region that are decreased in DKD vs. control (not just the peak immediately below the orange arrows). We understand that this may create some confusion and have removed the orange arrows. Each of these peaks in PCK1 meet the adjusted p-value threshold and log-fold-change threshold.

-I am unsure about the additional value of tCRE analysis, as from the text it is not quite evident what orthogonal information it really provides. The argument that PT_VCAM1-specific tCRE shows enhanced transcription of VCAM_1 is also not surprising. I am lost with this part

The tCRE analysis shows that 5-prime snRNA-seq datasets can capture information about cis-regulatory relationships and alternative promoter usage. Our dataset is unique in that we have both 5-prime snRNA-seq and snATAC-seq, which allowed us to show that a subset of open chromatin regions are also transcribed. This raises the possibility that transcription of cis-regulatory elements could provide evidence of enhancer activity. The PT_VCAM1 upstream tCRE was only recently-described and there were no known transcribed regulatory elements in publicly-available databases. We did not expect a priori that the 60kb upstream peak was actively transcribed.

-The comparison of PT_VCAM1 fractions between DKD and Control samples gives a p of 0.14, which indicates no statistically significant difference, the authors should not claim a “trend”. The more interesting question would be if the authors can compare the fraction of these cells to non-diabetic CKD samples. What is the authors’ take on specificity of the PT_VCAM1 population for DKD development, progression?

We have repeated the PT_VCAM1 fraction analysis between DKD and control samples using a non-parametric test and the results are statistically significant. We have updated the manuscript to reflect

the new approach. Although it would be of interest to compare DKD to non-diabetic CKD samples, we feel that these experiments would be better suited to a future study. PT_VCAM1 is likely important for disease progression in both DKD and non-diabetic CKD. There is a recent preprint by the Kidney Precision Medicine Project (KPMP) <https://www.biorxiv.org/content/10.1101/2021.07.28.454201v1> and in this manuscript they describe an “adaptive” or “maladaptive” population of proximal tubule cells that express *VCAM1*, *PROM1*, and other markers of injury that make them very similar to our PT_VCAM1 population. They associated an “adaptive” epithelial cell injury signature with poor clinical outcomes (see Figure 6 in preprint) such that patients with increased expression of these markers were more likely to experience a 40% or greater reduction in eGFR or progression to ESKD. The KPMP preprint grouped all CKD patients together, so it’s likely that PT_VCAM1 plays a role in DKD and non-diabetic CKD.

-No QC reported for CUT&RUN data. Please provide the QC.

We have included QC metrics for the CUT&RUN data from both the hTERT-RPTEC and bulk kidney experiments in the tables below. The first table is for GR and the second table is for the IgG isotype control. We have also incorporated these data into the manuscript.

Modality	QC	hTERT GR_rep1	hTERT GR_rep2	Kidney GR_rep1	Kidney GR_rep2	Kidney GR_rep3
Bowtie2 QC	Total reads	18673896	24620667	26797818	21358232	25923614
	Overall alignment rate	69.35%	80.64%	57.70%	43.47%	84.95%
Picard QC	PERCENT_DUPLICATION	0.293177	0.234621	0.24355	0.424802	0.428823

Modality	QC	hTERT IgG_rep1	hTERT IgG_rep2	Kidney IgG_rep1	Kidney IgG_rep2	Kidney IgG_rep3
Bowtie2 QC	Total reads	22670636	22869508	24459172	27088334	26940523
	Overall alignment rate	76.25%	56.54%	63.21%	55.74%	72.96%
Picard QC	PERCENT_DUPLICATION	0.277117	0.438572	0.276825	0.470888	0.422493

-Fig5E- the decreased accessibility of FKBP5 is not obvious in diabetic samples. Even in the promoter I see higher accessibility in diabetic samples.

Thank you for your careful attention detail. On review, some of the peaks that were marked as differentially accessible in the FKBP5 gene body met the adjusted p-value threshold, but did not meet the log-fold-change threshold in PCT. We have removed the orange arrows for those peaks and retained a single FKBP5 promoter peak that shows decreased accessibility in the diabetic proximal tubule (highlighted in red in the table). As you can see from the table below, there are 7 DAR that meet the adjusted pval threshold in 3 cell types. Two of the DAR meet the effect size threshold and they are in PCT and PST. Notably, the DAR in PST that meets the effect size threshold is also present in PCT where it does not meet the threshold. For all of the FKBP5 DAR that meet the adjusted p-value threshold, 4 of 7 intersect with differentially-methylated regions obtained from publicly-available databases of DKD and/or CKD vs. control kidney samples (see additional reviewer response below).

peak	avg_log2FC	p_val_adj	celltype	gene	Overlap DMR
chr6-35731123-35732534	-0.100718473	1.49E-09	PCT	FKBP5	YES
chr6-35722162-35722814	-0.08017022	6.99E-07	PCT	FKBP5	NO
chr6-35691074-35691822	-0.048799466	1.75E-05	PCT	FKBP5	YES
chr6-35583675-35584750	0.024291223	0.007438537	PCT	FKBP5	NO
chr6-35722162-35722814	-0.107324461	4.58E-06	PST	FKBP5	NO
chr6-35727329-35728485	-0.076816138	0.001853007	TAL1	FKBP5	YES
chr6-35601246-35602994	-0.075665971	0.006358263	TAL1	FKBP5	YES

The peak that appears to have higher accessibility (on the left side) is at the 3' end of the transcript. The promoter is on the right and the gene is transcribed right to left. We have added text to indicate the 5' and 3' ends of the transcript to make this more clear.

-The authors' rigidity of analytical approach seems questionable at times: E.g., in the scRNAseq results part, they talk about certain glucose pathway genes differentially expressed after showing enrichment for glucose and glucocorticoid pathways. Important to note that the genes the authors mention did not end up among the top DEGs in the first place, as the log fold changes are miniscule. This reviewer cannot help but get the impression that the authors – probably for the sake of presenting a coherent storyline – sometimes sacrifice objectivity of bottom-up data analysis in favor of a top-down approach, zooming in a lot on tiny fold changes between a single cell type of interest (in this case PT) when analyzing DEGs between DKD and control samples. A log2 FC of 0.19 in the case of NR3C1 seems somewhat underwhelming. Rather, it would be of interest what the log2 fold changes for this gene are in other tubular segments (PT_VCAM1, LOH, DCT, collecting duct, etc.) and compare them head-to-head. So far, I am not fully convinced that these changes are relevant to justify highlighting them (just because they fit the ATAC analysis, for that matter) or that they are even cell type-specific. Along those lines, while it is true that the PT is important for glucose reabsorption and metabolism, it needs to be pointed out that – especially given the meager log2FC for, e.g., SGLT2 or PCK1, most other (also tubular) cells in the kidney also feed on and produce and metabolize glucose, for which they import glucose via various transporters (GLUTs, SGLT1, etc.) and process with enzymes etc. A comprehensive analysis of glucose transport, production, and metabolism with focus on effect size and cell type specificity is warranted to exclude compensatory effects in other (tubular) cell types than PT that might even be larger than the presented effects in PT. Most importantly there is a lack of a clear message in here.

We appreciate the reviewers' comments, but we strongly feel that glucose metabolism and glucocorticoid signaling are critical pathways in DKD pathogenesis. Both the snATAC-seq and snRNA-seq datasets independently implicated both gluconeogenesis and glucocorticoid signaling in pathway analysis. We focused on the proximal tubule because this is the primary site for glucose reabsorption and the only cell type with the requisite enzymes for gluconeogenesis. As you can see in the plot below, no other cell types in the kidney express the rate-limiting enzyme, *PCK1*. Similarly, other segments of the kidney do not significantly contribute to glucose reabsorption or circulating glucose levels. SGLT1 is primarily expressed in PST and only reabsorbs 3% of glucose compared to 97% by SGLT2 in PCT (PMID: 30132032). In the snRNA-seq data, *PCK1* showed a 62% increase in expression in the proximal tubule.

This is the rate-limiting enzyme for gluconeogenesis and we would expect that a 62% increase is more than miniscule. In contrast, *SLC5A2*, which encodes the sodium glucose cotransporter, only increased 23%, and this is a key therapeutic target in the proximal tubule.

In regards to *NR3C1* expression, we observed increased expression in PT ($\log_2FC = 0.20$), TAL1 ($\log_2FC=0.34$), and TAL2 ($\log_2FC=0.27$). The PT and TAL are two segments of the nephron where GR can exert its effects on both metabolism and ion absorption. These are the same cell types where we observed changes in chromatin accessibility in *FKBP5*.

GLUT1 (*SLC2A1*, GLUTS) and GLUT2 (*SLC2A2*) are not detectable by snRNA-seq in the kidney.

Given that the proximal tubule is the primary site for glucose reabsorption and glucose production, we thought it would be an excellent cell type to focus on. Our message is depicted in figure 8 where we hypothesize that chromatin accessibility in the diabetic proximal tubule modulates cellular responsiveness to GR signaling to influence metabolic and inflammatory pathways.

Minor comments

-Why did you use 17 cycles for cDNA preparation of your single cell libraries? According to the protocol the maximum should be 13. It can seriously alter the validity of the results.

Targeted Cell Recovery	Low RNA Content Cells e.g., Primary Cells Total Cycles	High RNA Content Cells e.g., Cell Lines Total Cycles
500-2,000	16	14
2,001-6,000	14	12
6,001-10,000	13	11

We appreciate the reviewers comments and would like to note that the 10X user guide (see above) has a recommended range from 11-16 cycles depending on rna content and targeted cell recovery. In addition, the 10X protocols were developed for pbmc and are not always suitable for analyzing kidney tissue.

-The demultiplexing protocols for your snATAC-seq and snRNA-seq libraries should be mentioned.

We have updated the methods section to state that demultiplexing was done with bcl2fastq per standard procedures.

-Please provide the data QC both for the snATAC-seq and the snRNA-seq datasets.

We have provided additional QC plots and metrics in the supplementary material as requested (Supplemental Figures 10 and 11)

What statistical test was used for comparison? “There was a trend towards greater proportion of PT_VCAM1 in DKD samples compared to healthy controls (0.09 vs. 184 0.25, p = 0.14).”

We have updated the manuscript to state that a Wilcoxon rank sum test was used.

-How did you compare the fractions with students’ t test in this part? “There was a trend toward greater proportion of PT_VCAM1 in DKD compared to healthy controls (mean proportion 0.06 vs. 0.02, 241 Student’s t-Test p=0.051)”. You could use chi-square.

We have updated the manuscript to state that a Wilcoxon rank sum test was used.

-Please provide the UMAPs of both your dataset in terms of original clustering as well as representation of samples to show the batch effect.

We have provided additional QC plots and metrics in the supplementary material as requested.

-Ambient RNA is very important in snRNA-seq that could affect the results. Was the data corrected for ambient RNA if not why?

We did an ambient RNA correction with SoupX and found that there was very low RNA contamination with an estimated $\rho=0.04$. Ambient RNA correction did not alter the overall structure of the data (see plots above). For these reasons, we decided not to perform ambient RNA correction.

Fig. 2B bears almost no additional information to the text and could be omitted. Maybe certain genes could be highlighted to make it useful in some way and help focus?

We agree that Fig2B does not have much additional information and have removed it from the text.

Fig. 2F: hard to see any changes in the violin plots. Dot plots might be better suited.

We have added dot plots to accompany the violin plots.

It's transposase-accessible chromatin, not transpose-accessible chromatin

Thank you for pointing out this typo. We have changed the transpose-accessible typo to match other mentions of transposase-accessible chromatin in the manuscript.

Reviewer #2 (Remarks to the Author):

To identify the altered signaling pathways and transcription factors associated with diabetic kidney disease (DKD), the authors generated single nucleus RNA- sequencing (snRNA-seq) and assay for transposase accessible chromatin sequencing (snATAC-seq) on human kidney cortex from donors with and without DKD. The authors identified increased proportion of inflammatory VCAM1+ injured proximal tubule cells (PT_VCAM1) in DKD. snATAC seq data identified the cell-specific differentially accessible regions (DAR) in nephron in DKD. DAR enriched for glucocorticoid receptor (GR) motifs. The authors found that changes in chromatin accessibility are associated with decreased insulin receptor expression, increased gluconeogenesis, and decreased expression of the GR cytosolic chaperone, FKBP5, in the diabetic proximal tubule. Cleavage under targets and release using nuclease (CUT&RUN) of GR in control cortex tissue and renal proximal tubule epithelial cells (RPTEC) showed co-localization of DAR and GR binding sites. Moreover, the authors demonstrated that CRISPRi silencing of GR response elements (GRE) in the FKBP5 gene body reduced expression of FKBP5 (a negative regulator of GR) in RPTEC. In addition, the authors showed that genetic background might regulate chromatin accessibility and DKD progression. Moreover, the authors developed open-source tools for single-cell allele-specific analysis (SALSA) to model the effect of genetic background on gene expression.

Overall, the manuscript is well written, and the conclusions are derived from several state-of-art techniques and analyses and largely supported by the results. This is one of the first studies to perform integrative multimodal single nucleus sequencing in DKD kidney samples. The findings provide a better understanding of the single-cell landscape and genetic heritability in DKD. They are

also helpful to other researchers to identify the cell-specific changes in the gene of interest in their kidney datasets. I have outlined a number of aspects to be addressed to improve the paper.

We appreciate these positive comments.

General Comments:

1. Abstract: Can be more informative to highlight the major objectives and findings of the study and the key conclusions. The current Abstract provides technical details without providing the rationale or broad conclusions and clinical implications.

We appreciate the reviewer feedback and have revised the abstract to include more details surrounding the rationale, broad conclusions, and clinical implications.

2. The authors recently (Nat Commun 2021, Muto et al) used healthy adult human kidneys to perform sn-RNA-seq and sn_ATAC-seq. Presumably, some of those datasets are also included in the current study (ie the control samples). In this previous study, they also found a subpopulation of proximal tubule that had increased expression of VCAM1 (PT_VCAM1) as well as HAVCR1 (kidney injury molecule-1). So it appears these marker of tubular injury are enriched even in “healthy” kidney PCTs. Can the authors comment on this? Given the relatively small n’s, and without correction for various variables (including HbA1c, blood pressure, age, etc) could some of the changes be also attributed to one or more of these confounders?

This is an important point and we have expanded our discussion to incorporate some of these questions. There is indeed a subset of proximal tubule cells in non-diabetic control kidney that express *VCAM1* and *HAVCR1*. They are detectable by single cell sequencing and immunofluorescence where they have a heterogeneous staining pattern that may represent multiple subpopulations. In our earlier manuscript (Muto et al.), we estimated that this population comprises roughly 2% of total cells and 6% of proximal tubule cells. We hypothesize that a subset of cells fail to repair and adopt a pro-inflammatory senescent phenotype (ie PT_VCAM1). This model would be consistent with what we’ve observed in our mouse IRI studies (albeit on an accelerated timeline). The proportion of these cells increases with aging, CKD, DKD, and ischemic injury. Given that our “healthy” control samples are obtained from donors older than 50, it’s likely that they’ve already accumulated some aging-associated changes in the kidney. In the present study, PT_VCAM1 increased to ~7% of total cells and ~14% of proximal tubule cells in DKD. We hypothesize that DKD (and non-diabetic CKD in general) causes proximal tubule injury which alters its expression profile resulting in an increased proportion of PT_VCAM1. The age of our control samples and DKD samples was not statistically different, however, we cannot entirely exclude confounders like vascular disease, hemoglobin A1c, and other comorbidities. The emergence of the PT_VCAM1 cell state has been shown in a number of studies, including the recent KPMP preprint where they refer to it as an “adaptive” or “maladaptive” cell state <https://www.biorxiv.org/content/10.1101/2021.07.28.454201v1.full> . An interesting finding from the KPMP study was that an “adaptive” expression profile in the kidney predicted composite outcomes like eGFR decline and ESKD. In summary, we hypothesize that PT_VCAM1 is a marker of active and/or cumulative injury and inflammation. PT_VCAM1 is not specific to DKD, but what’s less clear is whether PT_VCAM1 behaves differently in DKD vs. non-diabetic CKD.

3. In Figure 3B – The authors noticed a significant proportion of tCRE in intronic (Figure 3B, 11847/37,698, 31%) and intergenic regions (Figure 3B, 1367/37,698, 3%) which they speculate may be representative of enhancer RNA (eRNA). In general, eRNAs are not easily captured by single cell sequencing technologies due to low or cell specific expression or rapid turnover. Please comment.

We agree that single cell sequencing has low sensitivity for detecting eRNA given their low expression and rapid turnover, but there are some unique advantages to our 5-prime snRNA-seq approach. The majority of eRNA are not polyadenylated (PMID: 32810208), which would make them extremely difficult to detect with the more commonly used 3-prime sequencing. 5-prime sequencing captures non-polyadenylated RNA, which can be analyzed using cap analysis of gene expression (CAGE) (PMID: 32124327). CAGE was originally developed for 5-prime bulk RNA-seq but has been adapted for 5-prime single cell RNA-seq. The major difficulty with CAGE is distinguishing alternative transcriptional start sites from artifacts. It's likely that some of the tCRE, especially in intronic regions, represent either alternative transcriptional start sites or artifacts (and not eRNA). We think that tCRE in intergenic regions may be particularly interesting because they're less likely to be an alternative TSS for a protein coding gene.

4. All Supplemental tables are missing from the manuscript (Will these be provided on request?).

All supplemental tables have now been provided.

5. Line no. 159-167: Provide the data describing in these lines, which might be helpful for readers.

This information is included in supplemental table 1.

6. In the Supplement Figure, provide the data for proportions of the cell's numbers identified in control and DKD samples for snATAC-sq and snRNA-seq.

This information is included in supplemental table 1.

7. In the Supplementary Fig, provide the data for ATP1B1 and KCNE1B, like figure 1E.

A figure for ATP1B1 has been added to the supplementary information to show differentially accessible regions in both the proximal tubule and TAL in the same locations. Interestingly, both of these DAR also coincide with GR binding sites by CUT&RUN and a differentially methylated region associated with DKD progression from a previously-published database. These data raise the possibility that changes in chromatin accessibility in the proximal tubule and TAL alter GR binding in the ATP1B1 promoter to modify sodium transport in DKD. The evidence for KCNE1B involvement is somewhat weaker and difficult to visualize due to the distance from the DAR to the gene body and we have omitted this gene from the text.

8. A similar figure like figure 2E can be generated and included in the supplementary figure for ALDOB, FBP1, and G6PC.

Figures for ALDOB, G6PC, and FBP1 have been added to the supplementary data (Supplemental Figures 7-9). Notably, ALDOB, FBP1, and G6PC all have differentially methylated regions (DMR) within their gene body or overlapping a CCAN.

9. The authors showed the RNA expression of GR and FKBP5 in the manuscript. Is the protein expression of GR and FKBP5 altered in control versus DKD samples? This could be tested by western blot/ immunohistochemistry analysis.

We appreciate the reviewers' comments, but human DKD samples are precious and very difficult to obtain and we are concerned that a western blot and/or IHC may not be able to show a 30-50% reduction in FKBP5 expression. Western blot and IHC are likely to be confounded by a mixture of multiple cell types and sample-to-sample heterogeneity, which is not easily normalized like snRNA-seq and snATAC-seq.

10. In lines 675-678, the authors hypothesize that DNA hypermethylation might be changed (hypermethylated) near FKBP5. Authors can provide DNA methylation data at the differential snATAC peaks in FKBP5. Authors can also check if publicly available data sets on DNA methylation in diabetes/ cardiovascular risk overlap with the differentially accessible regions (DAR) from snATAC-seq/GR peaks.

We appreciate the reviewers' comments, but we feel that methylation analysis is beyond the scope of this manuscript. The majority of methylation studies are carried out on dozens or even hundreds of samples and are not at single cell resolution. It's unlikely that our limited sample size would be able to identify any significant changes between control and DKD samples. To address this question, we have explored the literature to determine whether differential methylation has been previously-reported in diabetes and CKD in the same regions as our cell-specific DAR and GR CUT&RUN sites.

We extracted differentially methylated regions (DMR) from publicly-available datasets, including DKD (PMID:31165727, PMID:33933144, PMID:33144501) and CKD (PMID:24098934, PMID:24253122). Approximately 9% of our cell-specific DAR (that met both the adjusted p-value threshold and effect-size threshold) were located within a 1kb window of a DMR (N=120/1315, 9.1%). These DAR were spread across multiple cell types (see table below). FKBP5 contained multiple DMR; some of which overlapped the DAR described in our study. The PCT snATAC-seq peak located in the promoter region ~2kb upstream of the 5' FKBP5 TSS (Figure 5E: orange arrow, chr6:35731123-35732534) overlaps with both a GR CUT&RUN binding site and a DMR that showed increased methylation (fold-change=1.6) associated with end-stage kidney disease due to diabetes. This same study (PMID: 33933144) reported 6 additional DMR in the FKBP5 gene body. All of the FKBP5 DMR showed increased methylation and the majority of them overlapped with an snATAC-seq DAR that met the adjusted p-value threshold (but was below the effect-size threshold, see table above in reviewer response #1). FKBP5 was also identified as a top ranked marker with multiple CpGs showing differential methylation in CKD (PMID:24253112, Supplemental Table 2).

Overlapping DAR with DMR	
celltype	No.
ATL	5
DCT1	5
DCT2	4
PC	1
PCT	41
PST	18
PT_PROM1	3
PT_VCAM1	17
TAL1	12
TAL2	14
Total	120

We subsequently compared our GR CUT&RUN sites with the same set of DMR.

Approximately 6% of GR peaks in bulk kidney (N=269/4362, 6.1%) overlapped with a DMR in either DKD or CKD. The overlap between bulk GR CUT&RUN sites included 310 unique DMR. There were 6 DMR within or near the FKBP5 gene body that overlapped with a bulk GR CUT&RUN peak and all of these DMR showed increased methylation.

Approximately 6% of GR peaks in hTERT-RPTEC (N=1537/22517, 6.8%) overlapped with a DMR in either DKD or CKD. The overlap between hTERT-RPTEC GR CUT&RUN sites included 1554 unique DMR. There were 7 DMR within or near the FKBP5 gene body that overlapped with an hTERT-RPTEC GR CUT&RUN peak and all of these DMR showed increased methylation. The majority of these DMR are the same regions that overlap with bulk kidney GR CUT&RUN peaks and are associated with end-stage kidney disease due to diabetes.

We have included the results of our DMR comparison analysis in the supplementary material and added a section to the text to describe our results.

11. Authors can provide the list of anti-inflammatory effect genes as hypothesized in Figure 8.

We used the GR regulatory network genes obtained from msigdb (Gene Set: PID_REG_GR_PATHWAY) and intersected them with cell-specific degs to obtain a candidate list of GR-regulated genes that change expression in DKD. Many of these genes have well-established roles in inflammation (see table below)

gene
CDKN1A
CREBBP
EGR1
FKBP5

FOS
HSP90AA1
ICAM1
JUN
MAPK10
NCOA2
NFKB1
NR3C1
NR4A1
PBX1
PCK2
PRKACB
SGK1
SMARCA4
STAT1
STAT5A
STAT5B

Reviewer #3 (Remarks to the Author):

Summary: The authors present an analysis of alterations in single-nucleus chromatin availability and transcriptional signature in DKD vs. control samples. Key findings include a trend towards an increased proportion of VCAM1+ proximal tubule cells in DKD samples, cell-type specific ATAC peaks showing enrichment for heritability of kidney-related traits, enrichment of glucocorticoid and mineralocorticoid receptor motifs in genomic regions differentially accessible in DKD, and association of SNVs in a subset of ATAC peaks with alterations in expression of the the target genes. The manuscript represents a significant advance on previous work, providing to my knowledge the first single-nucleus ATAC-seq profiling of diabetic kidney disease samples, and is generally thoughtfully written, interesting, and clear.

We appreciate these positive comments.

Comments:

It would be interesting to know for several of the analyses how the reported trends are distributed across patients and between control vs. DKD patients. For example:

-How many cells of each type come from each patient for snRNA-seq and ATAC-seq datasets?

The distribution by cell type and sample for snRNA-seq and snATAC-seq are now included in supplemental table 1

-What is the distribution of PT_VCAM1 proportion across DKD and control patients for the snATAC-seq and snRNA-seq datasets?

-What is the distribution of average INSR expression in PT across DKD and control patients?

-For heterozygous SNVs that appear across multiple patients, how consistent are the estimated effects on gene expression across individuals?

This is an excellent question that gets at the problem of disentangling the effect of genotype from a multitude of confounders like age, sex, diabetes etc. We extracted the ATAC peaks for which gene expression was a significant predictor ($p < 0.05$) of allele-specific chromatin accessibility from our base model (gene expression as a fixed effect and donor as a random effect). Among the 7512 significant peak-gene combinations, the median between donor variation was 0.14 and the median absolute expression effect size was 0.01. In the base model, a minority of peak-gene combinations had an effect size that was larger than the between donor variation ($N=2615/7512$, 34%), suggesting that there is significant between donor variability. Some of this variability can be attributed to the effect of diabetes on gene expression, which we adjusted for in the full model. In the full model, there were 7,353 peak-gene combinations with a significant predicted effect of gene expression on allele-specific chromatin accessibility. The median between donor variation of these peak-gene combinations was 0.09, which suggests that adjustment for diabetes helps address some, but not all of the variability observed in the base model. The median absolute effect size of the full model was 0.012 in the absence of diabetes and 0.34 in the presence of diabetes. Together these data suggest that SNV may exhibit variable effects on gene expression in the presence and absence of diabetes and that the full model still contains significant between donor variability that may arise from the heterogeneity of DKD, or other variables like age, sex, and comorbidities.

In addition, we ran individual models for each of our samples to determine the reproducibility of effect size and direction across donors. There was a total of 10,495 nominally significant (Wald $p < 0.05$) peak-gene combinations present in two or more donors. Approximately two-thirds of these peak-gene combinations (7121/10495, 67%) had an effect size in the same direction despite the confounding

effects of age, sex, diabetes etc. Our model does not account for the type of SNV substitution or the location within the ATAC peak, so this is likely an additional source of variability between donors.

In summary, gene expression has a modest association with allele specific chromatin accessibility and exhibits significant donor variability. It's conceivable that a larger sample size would have more power to address these confounding variables.

Why is the number of GR binding sites specific to hTERT-RPTEC compared to IgG-stimulated control samples (22,539) so much larger than the number of GR binding sites identified in bulk kidney cortex from a healthy donor (4,362)? How do the location of the binding sites compare between hTERT-RPTEC and healthy donor kidney?

We hypothesize that hTERT-RPTEC has more GR binding sites because it is a more homogeneous sample composed of a single cell type and is stimulated with a saturating concentration of dexamethasone. As a result, the hTERT-RPTEC peaks are not diluted by other cell types and we have greater sensitivity for detecting peaks. In addition, bulk samples likely have reduced sensitivity due to preanalytic variables like ischemic time and tissue processing conditions. That being said, we believe that the bulk GR peaks we detected is likely an underestimate of the true number.

We compared the hTERT-RPTEC and bulk kidney GR CUT&RUN peaks to determine how many are shared between the datasets. Approximately one third (N=1289/4289, 30%) of the bulk GR peaks directly overlap an hTERT-RPTEC peak, which seems like a reasonable proportion corresponding to the expected fraction of PT.

Line 514-515: SALSA seems to be similar or perhaps extended from a method previously described by the authors in (Muto et al. Nat Commun. 2021 Apr 13;12(1):2190) and indeed the github page for SALSA suggests the previous manuscript as the citation. The original citation should be provided on lines 514-515 and the text should explain what has changed in the method from the previous work. (Is this a repackaging of the previous code to make it more usable, or has the method substantially changed from the previous approach)?

An early version of the genotyping and single cell allele-specific counting pipeline was developed for Muto et al. That manuscript used single cell allele-specific counts to model allele-specific expression using a tool called ASEP (PMID: 32392242). The SALSA workflow adds reference-based variant phasing, multithreading, command line tools, a docker container, and the generalized linear models for allele-specific chromatin accessibility described in this manuscript. The end result is a combination of repackaging previous code to make it more usable (eg. Multithreaded genotyping and user tutorials) and new methods (eg. Variant phasing, GLMM for allele-specific chromatin accessibility). We have updated the manuscript to describe the changes in more detail.

Minor comments:

Abstract line 37-38: The increased proportion of VCAM1+ cells does not pass a significance threshold in either snRNA-seq or sn-ATAC seq, so consider reporting it in the abstract as “a trend towards an increased proportion of VCAM1+ injured proximal tubule cells” as in the text.

As suggested by another reviewer, we now use a non-parametric test to compare the proportions between groups and the results are significant. We have added text to the discussion to indicate that this will need to be validated in future studies.

Line 151: The control patients cannot really be described as “healthy” as these samples are from tumor nephrectomy and deceased organ donors. Perhaps describe these patients as just “control patients.”

This is an excellent point and we have removed the “healthy” description from the text.

Line 178: It looks like it is Supplemental Figure 2 (not Supplemental Figure 1) that shows identification of cell types “based on increased chromatin accessibility within gene body and promoter regions of lineage specific markers.”

Thank you for your attention to detail. We have corrected this error in the text.

Line 184-5: Are these means of the cell type proportion across samples?

That is correct. We have updated the text to make this clarification.

Figure 1E caption: This is a very modest fold change (0.92; log-fold change -0.11). Presumably the significant p-value is possible because of the large number of cells analyzed. Is the direction of the fold change seen consistently across pairwise comparisons between the DKD patients and the healthy patients?

We appreciate the reviewer comments and recognize that the changes in chromatin accessibility are quite modest. As pointed out, the adjusted p-value for this particular region is 1.9×10^{-18} , which is reflected by the large number of PCT cells analyzed (N=22,821). We conducted all of the pairwise comparisons between control and DKD samples for this region to create the table below. The majority of pairwise comparisons estimate a reduction in chromatin accessibility for diabetes vs. control (blue fill = 28/35) and all of the pairwise comparisons that achieve statistical significance (red text = 15/35) show reduced chromatin accessibility.

Figure 1F caption: The text appears to be truncated--the caption ends with “(fold-”. Presumably the fold-change and p-value would be provided in the rest of the caption.

We have corrected the truncated figure legend.

Lines 306, 309: The wrong supplemental table appears to be referenced here; Supplemental Table 7 has the comparison between PT and PT_VCAM markers, not the cell type specific tCRE.

We appreciate the reviewers’ attention to detail and have corrected the reference.

Line 338: Was this sample obtained from one of the 6 control donors from the snRNA-seq analysis? Consider providing the anonymized id (e.g “Control1”) so that the reader can cross-reference to the metadata in Supp Table 1 (or provide anonymized metadata for this donor).

The bulk kidney GR analyses were obtained from separate donors and were not analyzed by snRNA or snATAC. We have added detail to the methods section to clarify this point. Additionally, supplemental table 1 now contains all of the metadata for each of the snRNA and snATAC libraries.

Figure 4A legend: What does “distance between the peak and its two neighboring regions” mean? Is this minimal distance to a genic region?

The distance between the peak and its two neighboring regions refers to the minimum distance between the peak and the next two closest ATAC peaks. If ATAC peaks are close together then the minimum distance between neighboring peaks goes down and the height of the log-transformed y axis

increases. This helps to visualize genomics regions with clusters of peaks from regions that have very few peaks. We have updated the legend to change “two neighboring regions” to “two neighboring peaks”.

Figure 6 appears to be lower-resolution than other figures in the paper.

We have included a high resolution version of this figure with the current submission.

Lines 551, 560, 563, 572, 623, 625: should these be “nominally significant”/ “nominal p-value threshold” since these peak-gene combination counts appear to be without multiple hypothesis testing correction?

That is correct. We have updated the manuscript to clarify that these are nominally significant peak-gene combinations.

Line 557: “this translates to the typical ATAC peak being 1.10 times more likely to contain an alternate allele in the base model:” This seems like a very modest increase in the probability of containing an alternate allele, and I am surprised that such a small allele bias would lead to a 10% change in gene expression. Is the text correct as stated?

That is the correct interpretation. An important caveat for the base model is that it is not adjusted for diabetes or the interaction between diabetes and gene expression. Some of the change in gene expression can be attributed to those variables so the magnitude of change due to the allele-specific effect is likely much smaller. We have added some additional text to the manuscript to clarify these points.

Figure 7A, B: Although no cell types pass the significance threshold for microalbuminuria, the color of all of the bars is different from the color of the the non-significant bars in the other plots.

We have changed the colors of the bars to match the other plots.

Line 628: I don’t think that the acronym for generalized linear mixed model (GLMM) is defined anywhere in the manuscript.

We have updated the text to spell out generalized linear mixed model in the figure legend.

I would suggest including in the supplemental materials an index of the supplemental tables with a brief description of the content of each table.

We have added an appendix for the supplemental tables as suggested.

REVIEWERS' COMMENTS

Reviewer #1 (Remarks to the Author):

I have no additional comments

Reviewer #2 (Remarks to the Author):

Overall, the authors have addressed my comments satisfactorily and updated the manuscript accordingly. A few issues remain.

1) Figure 4B - The text in the results and in the Legend for this Upset plot is far from adequate. It is not clear what is meant by "For example, there were GR binding sites in ATAC peaks unique to the proximal tubule (N=64, 284 Figure 4B), unique to the distal nephron (N=15, Figure 4B), and shared between the proximal tubule and 285 distal nephron (N=60, Figure 4B). Similarly, there were GR binding sites unique to lymphocytes and 286 shared between the proximal tubule and lymphocytes." Please also explain where lymphocytes and distal nephron data is shown. More detail is also needed to describe the bar graph with black bars on the right.

2) There are several elegant and state-of-the-art multimodal analyses and inferred conclusions in the manuscript. The tools developed and datasets are helpful to the community. In particular the data regarding GR is quite novel. To further enhance the outcome, the authors could provide a few sentences in the Discussion about the translational implications for DKD. What major pathway(s) were uncovered from this large scale profiling that were not known before and could therefore be exploited for adjunct therapy?

Reviewer #3 (Remarks to the Author):

The authors have satisfactorily addressed the questions that I raised. I find some of the additional data presented in the response to reviewers (for example, the comparison of VCAM1 proportion across DKD and control patients in scRNA-seq and scATAC-seq datasets, comparison of INSR expression in proximal tubule across patients, analysis of between-donor variation for heterozygous SNPs across patients, pairwise comparison of fold change between DKD patients and healthy patients in the INSR region) to be helpful for interpreting the results in the manuscript and would encourage the authors to include these analyses in the manuscript, perhaps as supplemental figures or notes.

Reviewer #1 (Remarks to the Author):

I have no additional comments

Reviewer #2 (Remarks to the Author):

Overall, the authors have addressed my comments satisfactorily and updated the manuscript accordingly. A few issues remain.

1) Figure 4B - The text in the results and in the Legend for this Upset plot is far from adequate. It is not clear what is meant by “For example, there were GR binding sites in ATAC peaks unique to the proximal tubule (N=64, 284 Figure 4B), unique to the distal nephron (N=15, Figure 4B), and shared between the proximal tubule and 285 distal nephron (N=60, Figure 4B). Similarly, there were GR binding sites unique to lymphocytes and 286 shared between the proximal tubule and lymphocytes.” Please also explain where lymphocytes and distal nephron data is shown. More detail is also needed to describe the bar graph with black bars on the right.

- We appreciate the reviewers comments and have now added additional detail to the legend for Figure 4B. Each column of the Upset plot in 4B indicates a group of cell types where group membership is indicated by the solid black circles. For example, the far left column has GR, PCT, ENDO, BCELL, and TCELL. This is the group we are referring to when we said that there are GR binding sites shared between the proximal tubule and lymphocytes. Each group of cell types denotes a unique set of ATAC peaks (all with a GR binding site) that are not seen in the other groups. The number of unique ATAC peaks in each group is the solid black bar on the top indicated by the ‘Intersection size’ axis. The GR binding sites that are unique to lymphocytes are shown in three separate columns. In the center of the plot, the 13th column from the left shows GR binding sites shared between BCELL and TCELL (n=15). On the right-side of the plot in the 2nd and 3rd to last vertical columns there are two groups that uniquely indicate ‘BCELL’ or ‘TCELL’. The GR binding sites that are unique to the proximal tubule are shown in the 2nd column from the left, which has a group consisting of PCT, PST, and PT_VCAM1. The GR binding sites that are unique to the distal nephron are shown in multiple vertical columns. From left-to-right, these groups consist of ‘DCT2 and PC’ in the 12th column, DCT1 in the 18th column, DCT2 in the 19th column, PC in the 20th column, and ICB in the 21st column. The far right column has a single solid circle over GR, which means that this group does not contain any GR binding sites in a cell-specific ATAC peak. The horizontal bars on the right side of the group indicate the total number of GR binding sites within each cell type across all of the groupings. If you travel left-to-right across a row and find all of the groups that a cell type belongs to (solid black circles) and then add up all of the intersection sizes, you will arrive at the ‘Set size’ on the right side of the plot.

2) There are several elegant and state-of-the-art multimodal analyses and inferred conclusions in the manuscript. The tools developed and datasets are helpful to the community. In particular the data regarding GR is quite novel. To further enhance the outcome, the authors could provide a few sentences in the Discussion about the translational implications for DKD. What major pathway(s) were uncovered

from this large scale profiling that were not known before and could therefore be exploited for adjunct therapy?

- We thank the reviewer for their positive comments and are hopeful that our work may be of use for the field. We hypothesize that targeting GR signaling in the proximal tubule to decrease GR-mediated gluconeogenesis may help to better control hyperglycemia, particularly during fasting. This approach might be useful in combination with SGLT2i, which have been shown to increase gluconeogenesis. We have added the following statement to our discussion:

“Targeting GR signaling in the proximal tubule may help to decrease GR-mediated gluconeogenesis and improve glycemic control, particularly during fasting. SGLT2i have been shown to increase gluconeogenesis, which raises the possibility that GR inhibition may be useful as a combination therapy, however, further studies will be needed to evaluate this hypothesis”

Reviewer #3 (Remarks to the Author):

The authors have satisfactorily addressed the questions that I raised. I find some of the additional data presented in the response to reviewers (for example, the comparison of VCAM1 proportion across DKD and control patients in scRNA-seq and scATAC-seq datasets, comparison of INSR expression in proximal tubule across patients, analysis of between-donor variation for heterozygous SNPs across patients, pairwise comparison of fold change between DKD patients and healthy patients in the INSR region) to be helpful for interpreting the results in the manuscript and would encourage the authors to include these analyses in the manuscript, perhaps as supplemental figures or notes.

- **We have added a supplemental figure (Supplemental Figure 4 in updated MS) to reflect the proportion of PT_VCAM1 by donor in both the snRNA-seq and snATAC-seq datasets.**
- **We have added a supplemental figure (Supplemental Figure 5 in updated MS) to show all pairwise comparisons for the INSR DAR in the proximal tubule accompanied by average INSR expression across donors.**
- **We have added to our results section to discuss the between-donor variation results for heterozygous SNV.**